# How alkaline compounds control atmospheric aerosol particle acidity

Vlassis A. Karydis[1,2*], Alexandra P. Tsimpidi[1,2,3], Andrea Pozzer[1,4], and Jos Lelieveld[1,5]

[1] Max Planck Institute for Chemistry, Atmospheric Chemistry Dept., Mainz, 55128, Germany.

[2] Forschungszentrum Jülich, Inst. for Energy and Climate Research, IEK-8, Jülich, 52425, Germany.

[3] National Observatory of Athens, Inst. for Environmental Research and Sustainable Development, Athens, 15236, Greece.

[4] International Centre for Theoretical Physics, Trieste, 34151, Italy

[5] The Cyprus Institute, Climate and Atmosphere Research Center Nicosia, 1645, Cyprus.

*Correspondence to*: Vlassis A. Karydis (v.karydis@fz-juelich.de)

**Abstract.** The acidity of atmospheric particulate matter regulates its mass, composition and toxicity, and has important consequences for public health, ecosystems and climate. Despite these broad impacts, the global distribution and evolution of aerosol particle acidity are unknown. We used the comprehensive atmospheric multiphase chemistry – climate model EMAC to investigate the main factors that control aerosol particle acidity and uncovered remarkable variability and unexpected trends during the past 50 years in different parts of the world. Our simulations revealed that the aerosol acidity trend is strongly related to changes in the phase partitioning of nitric acid, production of sulfate in aqueous aerosols, and the aerosol hygroscopicity. It is remarkable that the aerosol hygroscopicity (kappa) has increased in many regions following the aerosol particle pH. Overall, we find that alkaline compounds, notably ammonium, and to a lesser extent crustal cations, regulate the aerosol particle pH on a global scale. Given the importance of aerosol particles for the atmospheric energy budget, cloud formation, pollutant deposition and public health, alkaline species hold the key to control strategies for air quality and climate change.

## 1. Introduction

Aerosol particle acidity is a central property of atmospheric particulates that influence clouds, climate and air quality, including impacts on human health (Raizenne et al., 1996;Lelieveld et al., 2015). It affects the partitioning of semi-volatile acids between the gas and particle phases (Guo et al., 2016;Guo et al., 2017;Guo et al., 2018;Nenes et al., 2020), secondary organic aerosol (SOA) formation (Xu et al., 2015;Marais et al., 2016), the solubility of trace metals in aerosol particles (Oakes et al., 2012), associated with their toxicity (Fang et al., 2017) and nutrient capacity (Jickells et al., 2005), the activation of halogens that act as oxidants (Saiz-Lopez and von Glasow, 2012), the conversion of sulfur dioxide (Seinfeld and Pandis, 2006;Cheng et al., 2016), the particle hygroscopic growth and lifetime (Metzger et al., 2006;Abdelkader et al., 2015;Karydis et al., 2017), and

atmospheric corrosivity (Leygraf et al., 2016). Direct measurement of aerosol particle acidity is difficult and associated with
much uncertainty, being dependent on filter sampling and the $H^+$ molality in the aqueous extract, which is sensitive to artifacts
(Pathak et al., 2004). Therefore, particle pH, a commonly used acidity metric of aqueous aerosols, is typically inferred by
proxy techniques (Hennigan et al., 2015;Pye et al., 2020). Two of the most common are the ion balance and the molar ratio
methods. These methods do not consider the effects of aerosol water and multiphase interactions with gas phase species as
well as the partial dissociation of acids (Hennigan et al., 2015). The simultaneous measurement of gas phase species can
improve aerosol particle pH estimates by accounting for the phase partitioning of semi-volatile species (e.g., $NH_3$, $HNO_3$).
However, the accuracy of this approach relies on the availability of information on these species in both the gas and particle
phase, being scant in most cases.
The most reliable estimates of pH are obtained with thermodynamic equilibrium models, although the accuracy can be limited
by not accounting for all ionic species. For example, most atmospheric chemistry models do not consider crustal elements
(e.g., $Ca^{2+}$, $Mg^{2+}$, $K^+$) and $Na^+$ in sea salt. These species affect the ion balance by influencing the phase partitioning of nitrate
and ammonium, especially in areas where aeolian dust is abundant (Karydis et al., 2016). Here we present 50-year global
acidity trends of fine particulate matter (i.e. with a diameter < 2.5 μm) by employing the EMAC chemistry – climate model
(Jöckel et al., 2010). The pH calculations are performed online with the ISORROPIA II thermodynamic equilibrium model
(Fountoukis and Nenes, 2007).
**2. Results and Discussion**
**2.1 Global variability of aerosol particle acidity**
Figure 1 shows the modeled near-surface distribution of fine aerosol particle acidity for the 2010-2015 period. We find
predominantly acidic particles over the anthropogenically influenced regions in the northern hemisphere and the tropical
biomass burning zones, and mostly alkaline particles over deserts and oceans, especially over the southern oceans. The pH
typically ranges from 4.0 to 6.7 (5.3 on average) over the western USA since it is affected by crustal cations from the
surrounding deserts. Polluted areas located downwind of crustal sources are of special interest since the pH calculations can
be sensitive to the aerosol state assumption (see section 4.3). Over Pasadena, the base case model using the stable state mode
estimates a mean pH of 5.9 units, while the sensitivity simulation with only liquid particles results in 2.7 pH units (equal to
Guo et al. (2017) estimations by using the metastable assumption; Table A1). Our sensitivity analysis revealed that the aerosol
state itself is not affected by the state assumption since both stable and metastable predict the same amount of water in the
aerosol. Differences in the calculated pH can be due to the high concentrations of calcium from the Great Basin Desert which
results in the precipitation of high amounts of $CaSO_4$, lowering the particle acidity (but without affecting the water activity
since $CaSO_4$ is insoluble and does not contribute to the MDRH depression). It is worth mentioning that calcium was not
included in the Guo et al. (2017) study which helps explain the differences in the observed and simulated aerosol particle
acidity. The simulated particle-phase fraction of nitrate over Pasadena is 40% using the stable state assumption and 32% using
the metastable assumption, compared to the observationally derived 51%. Over Europe, the pH ranges from 2.6 to 6.7 (3.9 on
average). Observational estimates of aerosol particle pH from the Po Valley (Squizzato et al., 2013;Masiol et al., 2020) and
Cabauw (Guo et al., 2018) support the relatively low acidity of fine aerosols over Europe (Table A1). Model calculations
compare well with observational estimates from Cabauw, however, result in higher pH (~1 unit) compared to values from Po
Valley (estimated by using the E-AIM model). Over East Asia the average pH is 4.7, ranging from 2.6 to 7.4. Relatively high
pH are found over regions where anthropogenic aerosols are mixed with aeolian dust, e.g., from the Gobi Desert, which
decrease the acidity (e.g., ~6 pH units over Hohhot, which agrees well with the estimations of Wang et al. (2019a)). The
relatively low pH in large parts of Asia is explained by strong $SO_2$ emissions and associated sulfate, which have increased
strongly in the past decades (e.g., over Guangzhou, supported by estimations of Jia et al. (2018)). Estimates of unrealistically
high aerosol particle acidity can result from omitting the gas phase concentrations of semi-volatile ions from the pH
calculations (e.g., estimates over Hong Kong (Yao et al., 2007;Xue et al., 2011), Singapore (Behera et al., 2013) and Shanghai
(Pathak et al., 2009); Table A1). At the same time, $SO_2$ emissions have decreased over Europe and USA, and recently in China.
However, aerosol particles over the eastern USA have remained acidic, with an average pH of 3.0 until recently, corroborating
the findings of Weber et al. (2016) and Lawal et al. (2018) that aerosol particle acidity over this region is less sensitive to $SO_2$
than to $NH_3$ emissions.
The aerosol particle pH over polluted northern hemispheric mid-latitudes (e.g., over East Asia) and the northern extratropical
oceans exhibits a clear seasonal pattern with lower values during boreal summer and higher ones during winter, driven by the
availability of ammonium and by the aerosol water content (Fig. 2). This is evident from both our model calculations and from
observational estimates mostly in heavily populated areas such as the Po Valley (Squizzato et al., 2013), Beijing (Tan et al.,
2018), and Tianjin (Shi et al., 2017), and to a lesser extent over areas strongly affected by aeolian dust (e.g., Hohhot; Wang et
al., 2019b) (Table A1). Over tropical regions, fine particulates have a pH between 3.2 and 7.4, being strongly influenced by
pyrogenic potassium, i.e., from widespread biomass burning (Metzger et al., 2006), and a high aerosol water content.
Observational estimates from Sao Paulo support these high pH values (Vieira-Filho et al., 2016), albeit with 1 unit bias mainly
related to the use of the E-AIM model. Over deserts, aerosol particles are relatively alkaline, with a pH up to 7.4. Aerosol
particles in the marine environment tend to be alkaline also, with a pH up to 7.4 over the southern oceans. Observational
estimates report highly acidic aerosol particles over the southern oceans due to the lack of gas phase input for the pH
calculations (Dall'Osto et al., 2019). Over the Arctic and the northern Atlantic and Pacific Oceans, aerosol particle acidity is
significantly enhanced by strong sulfur emissions from international shipping and pollution transport from industrialized areas
(Fig. 1). The pH over the northern extratropical oceans and the Arctic ranges from 2.0 to 7.0 with an average of about 5.2. The
annual cycle of aerosol particle acidity over these regions is strongly influenced by anthropogenic pollution, being relatively
high during boreal summer. Over the Antarctic, aerosol particle pH ranges from 4.5 to 7.0 and follows a clear seasonal pattern
(Fig. 2).

## 2.2 Temporal evolution of aerosol particle acidity

Figure 1 and Table 1 present the aerosol particle pH over the period 1970-2020. We investigated the impacts of alkaline species by omitting the emissions of ammonia and mineral cations in two sensitivity simulations.

### 2.2.1 Europe

Over Europe, the pH has increased strongly from about 2.8 during the 1970s to 3.9 recently. Especially during the 1990s $NH_3$ emissions over Europe increased significantly by 14%, while at the same time NOx and $SO_2$ emissions decreased by 13% and 49%, respectively. While this trend has continued in the past decade, pH changes slowed because the sulfate and nitrate decreases have been compensated through volatilization of ammonia from the particles. In addition, the recently increasing cation/anion ratio is accompanied by a reduction of aerosol water, preventing a significant decrease of the aerosol particle acidity (Fig. S1). Overall, the increase of aerosol particle pH by more than 1 unit during the last 50 years had a significant impact on the gas-particle partitioning of semi-volatile acids, e.g., nitric acid, since their dissociation into ions enhances their solubility (Nah et al., 2018). Here, the fraction of nitrate in the particle phase relative to total nitrate (gas plus particle) has increased from ~70% to 85% (Fig. 3). The increase in aerosol particle pH has been accompanied by an increase in aerosol kappa hygroscopicity (Fig. 4). After the substantial reduction of $SO_2$ emissions, sulfate salts (e.g., ammonium sulfate with kappa=0.53) are replaced by more hygroscopic nitrate salts (e.g., ammonium nitrate with kappa=0.67) in the aerosol composition. In addition, the decrease of organic compound emissions during the last 50 years contributed to the increase of the aerosol hygroscopicity. Our sensitivity simulations reveal that aerosol particle acidity over Europe is highly sensitive to $NH_3$ emissions. Despite the decline of both $SO_2$ and NOx during the past decades, the aerosol particle would have remained highly acidic (pH ~1) in the absence of $NH_3$.

### 2.2.2 North America

Over North America, aerosol particle acidity also decreased with $SO_2$ and NOx emissions. Nevertheless, these emissions are still relatively strong in the eastern USA (5 times higher than in the western USA) resulting in very acidic aerosols, with a pH ranging from 2.2 in 1971 to 3.3 recently (Figs. 1 and S1). Such acidic conditions promote the dissolution of metals (e.g., Fe, Mn, Cu) in ambient particles (Fang et al., 2017). Soluble transition metals in atmospheric aerosols have been linked to adverse health impacts since they generate reactive oxygen species, leading to oxidative stress and increased toxicity of fine particulate matter (Fang et al., 2017;Park et al., 2018). Since the solubility of transition metals increases exponentially below a pH of 3, the decrease of aerosol particle acidity over the eastern USA reported here suggests that the particles have become substantially less toxic in the past few decades. Similar to Europe, the increasing pH has resulted in a growing aerosol particle nitrate fraction from ~50% during the 1970s to 65% recently (Fig. 3), and to a strong increase of aerosol hygroscopicity by ~0.15 units at the cloud base (Fig. 4). The role of $NH_3$ is critically important; without it the aerosol particle pH over the eastern USA would be

close to zero. Over the western USA, the aerosol particle pH is higher (~5), being affected by aeolian dust from the Great
Basin Desert, although $NH_3$ is still the most important alkaline buffer.

### 2.2.3 East and South Asia

In Asia, $SO_2$ and NOx emissions have increased drastically since 1970. However, the simultaneous increase of $NH_3$ emissions
along with the presence of mineral dust from the surrounding deserts (i.e., Gobi, Taklimakan, Thar) decelerated the increase
of aerosol particle acidity. Over East Asia, the aerosol particle pH decreased from about 5.3 during the 1970s to 4.5 in 2010.
This change in aerosol particle acidity has affected the predominant pathway of sulfate formation through aerosol aqueous
phase chemistry. Under acidic conditions, $SO_2$ is mainly oxidized by transition metal ions, while at pH > 5 the oxidation by
$O_3$ and $NO_2$ predominates (Cheng et al., 2016). Therefore, the decrease of pH during the last 50 years, even though being
relatively modest, was sufficient to turn-off sulfate production from $O_3$ oxidation (Fig. 5). At the same time, the increased
aerosol particle acidity hinders the partitioning of nitric acid to the particle phase, reducing the aerosol nitrate fraction from
90% to 80% (Fig. 3). Remarkably, the aerosol hygroscopicity has increased from ~0.3 in the 1970s to 0.45 recently (Fig. 4),
revealing a reverse development compared to Europe and the USA. Here, the fraction of mineral dust in the aerosol is higher;
therefore, the particles gained hygroscopicity by the acquired pollution solutes. Recently, the $SO_2$ emissions have dropped and
the NOx emission increase has slowed in East Asia, while $SO_2$ emissions are soaring in South Asia. $SO_2$ emission trends since
2007 have been so drastic that inventories and scenarios tend to overestimate the emitted $SO_2$. Satellite observations indicate
that India has recently overtaken China as the world largest emitter of $SO_2$ (Li et al., 2017). Following the satellite observations,
we implemented the significant $SO_2$ reduction trends into our model (Fig. S2). Surprisingly, the effect only becomes noticable
over East Asia after 2016, when the aerosol particle pH started increasing by about 0.3 units, while we do not find any change
over South Asia. This corroborates the strong buffering that we found over other regions such as Europe. Fig. 1 shows that
$NH_3$ has been the major buffer, supporting the recent findings of Zheng et al. (2020) that the acid-base pair of $NH_4^+/NH_3$
provides the largest buffering capacity over East and South Asia. However, we also found that in East Asia and to a lesser
extent in South Asia crustal elements, not considered in the study of Zheng et al. (2020), have contributed significantly on
maintaining a mean pH of 4.5 – 5 in the past decade (Fig. 1). Calcium is the major crustal component of dust from the Gobi
and Taklimakan deserts (Karydis et al., 2016) and unlike other crustal compounds it can react with sulfate ions and form
insoluble $CaSO_4$, which precipitates out of the aerosol aqueous phase. This interaction reduces the aqueous sulfate and thus
the aerosol particle acidity.

### 2.2.4 Tropical forests, Middle East

Over tropical forests, aerosol particles are typically not very acidic with pH values >4. Note that organic acids were not
included in the aerosol particle pH calculations, however, their contribution to the total ionic load is small (Andreae et al.,
1988;Falkovich et al., 2005), and aerosol particle acidity can be attributed to inorganic acids. Over the Amazon and Congo
basins, the aerosol particle pH remained around 5 since 1970. The Southeast Asian forest atmosphere is affected by pollution

from mainland Asia, and the aerosol particle pH decreased to around 4 recently. This pH drop has enhanced SOA formation from isoprene, since under low-NOx conditions (typical over rainforests) the presence of acidifying sulfate increases the reactive uptake of epoxydiols (Xu et al., 2015;Surratt et al., 2010). Nevertheless, NH$_3$ emissions provide a remarkably strong buffer over all three tropical regions while mineral dust cations are also important over the Amazon and Congo forests. Further, the Middle East is affected by strong anthropogenic (fossil fuel related) and natural (aeolian dust) aerosol sources. Due to the high abundance of mineral dust, the pH has remained close to 7. Without crustal cations, the pH would drop to about 4. Despite the omnipresence of alkaline species from the surrounding deserts, NH$_3$ still plays a central role in controlling the acidification of mineral dust aerosols, which can affect their hygroscopic growth and hence their climate forcing (Klingmuller et al., 2019;Klingmüller et al., 2020).

## 2.2.5 Oceans

Over the Arctic and northern extra-tropical oceans, aerosol particle acidity is strongly affected by pollution transport from the urban-industrial mid-latitudes. The Arctic aerosol particle pH is highly variable, remaining relatively low up to 1990 (~4.2), after which it increased to about 5.2. Crustal cations are found to play a significant role lowering the aerosol particle acidity. Over the northern extra-tropical oceans, aerosol particle pH has remained relatively constant (~4.8). NH$_3$ provides an important alkaline buffer, and without it the aerosol particle pH would have been below 3. NH$_3$ is also proved to be important over the tropical and southern extra-tropical oceans, where a noticeable increase in aerosol particle acidity occurred after June 1991, when the eruption of Mount Pinatubo in the Philippines released ~20 million tons of SO$_2$ into the stratosphere (McCormick et al., 1995). The impact of Pinatubo sulfate, after returning to the troposphere, on aerosol particle acidity is mostly evident over Antarctica, where the pH dropped by 2 units, as the stratospheric circulation is strongest in the winter hemisphere. Over Antarctica concentrations of dust and especially of NH$_3$ are very low, and Fig. 1 illustrates that only in this pristine environment the large Pinatubo anomaly could overwhelm the buffering by alkaline species. Except after Pinatubo, the pH has remained nearly constant at 5.8 over Antarctica and about 5.5 in the tropics and 6.8 in the southern extra-tropics.

## 3. Conclusions

We find that over Europe and North America the aerosol particle acidity decreased strongly in the past few decades resulting in substantially less toxic and more hygroscopic aerosols. At the same time, the particle acidity over Asia has decreased, even though the increase of NH$_3$ emissions and the presence of mineral dust decelerated the change in the aerosol pH. The inevitable decrease of the aerosol particle pH hindered the partitioning of nitric acid into the particulate phase and the sulfate production in the aerosol aqueous phase; however, the aerosol hygroscopicity increased over Asia following a reverse correlation with the particle pH. Overall, the aerosol particle pH is generally well-buffered by alkaline compounds, notably NH$_3$ and in some areas crustal elements. NH$_3$ is found to supply remarkable buffering capacity on a global scale, from the polluted continents to the remote oceans. In the absence of NH$_3$, aerosol particles would be highly (to extremely) acidic in most of the world. Therefore,

potential future changes in NH$_3$ are critically important in this respect. Agriculture is the main NH$_3$ source and a controlling
factor in fine particle concentrations and health impacts in some areas (e.g., Europe) (Pozzer et al., 2017). The control of
agricultural ammonia emissions must therefore be accompanied by very strong reductions of SO$_2$ and NOx to avoid that aerosol
particles become highly acidic with implications for human health (aerosol toxicity), ecosystems (acid deposition and nutrient
availability), clouds and climate (aerosol hygroscopicity).

## 4. Appendix A: Materials and Methods

### 4.1 Aerosol-chemistry-climate model

We used the ECHAM5/MESSy Atmospheric Chemistry (EMAC) model, which is a numerical chemistry and climate
simulation system that describes lower and middle atmosphere processes (Jöckel et al., 2006). EMAC uses the Modular Earth
Submodel System (MESSy2) (Jöckel et al., 2010) to link the different sub-models with an atmospheric dynamical core, being
an updated version of the 5th generation European Centre - Hamburg general circulation model (ECHAM5) (Roeckner et al.,
2006). EMAC has been extensively described and evaluated against in situ observations and satellite retrievals to compute
particulate matter concentrations and composition, aerosol optical depth, acid deposition, gas phase mixing ratios, cloud
properties, and meteorological parameters (Karydis et al., 2016;Pozzer et al., 2012;Tsimpidi et al., 2016;Karydis et al.,
2017;Bacer et al., 2018). The spectral resolution of EMAC used in this study is T63L31, corresponding to a horizontal grid
resolution of approximately 1.9$^{\circ}$x1.9$^{\circ}$ and 31 vertical layers extending up to 10 hPa (i.e., 25 km) from the surface. The presented
model simulations encompass the 50-year period 1970-2020.
EMAC calculates fields of gas phase species online through the Module Efficiently Calculating the Chemistry of the
Atmosphere (MECCA) Submodel (Sander et al., 2019). MECCA calculates the concentration of a range of gases, including
aerosol precursor species (e.g. SO$_2$, NH$_3$, NO$_x$, DMS, H$_2$SO$_4$ and DMSO) and the major oxidant species (e.g. OH, H$_2$O$_2$, NO$_3$,
and O$_3$). Aerosol microphysics are calculated by the Global Modal-aerosol eXtension (GMXe) module (Pringle et al., 2010).
The organic aerosol formation and atmospheric evolution are calculated by the ORACLE Submodel (Tsimpidi et al., 2014,
2018). The aerosol size distribution is described by seven lognormal modes: four hydrophilic modes that cover the aerosol size
spectrum of nucleation, Aitken, accumulation and coarse modes, and three hydrophobic modes that cover the same size range
except nucleation. The aerosol composition within each size mode is uniform (internally mixed), however, it varies between
modes (externally mixed). Each mode is defined in terms of total number concentration, number mean radius, and geometric
standard deviation (Pringle et al., 2010). The removal of gas and aerosol species through wet and dry deposition is calculated
within the SCAV (Tost et al., 2006) and DRYDEP (Kerkweg et al., 2006) submodels, respectively. The sedimentation of
aerosols is calculated within the SEDI submodel (Kerkweg et al., 2006). The cloud cover, microphysics and precipitation of
large scale clouds is calculated by the CLOUD Submodel (Roeckner et al., 2006) which uses a two-moment stratiform
microphysical scheme (Lohmann and Ferrachat, 2010), and describes liquid droplet (Karydis et al., 2017) and ice crystal (Bacer
et al., 2018) formation by accounting for the aerosol physicochemical properties. The effective hygroscopicity parameter κ is
used to describe the influence of chemical composition on the cloud condensation nuclei (CCN) activity of atmospheric
aerosols. κ is calculated using the mixing rule of Petters and Kreidenweis (Petters and Kreidenweis, 2007) and the individual
κ parameter values for each inorganic salt (Petters and Kreidenweis, 2007;Sullivan et al., 2009). Organic aerosol species are
assumed to have a constant hygroscopicity kappa parameter of 0.14 while bulk mineral dust and black carbon are assumed to
have zero hygroscopicity.

## 4.2 Emissions

The vertically distributed (Pozzer et al., 2009) CMIP5 RCP8.5 emission inventory (van Vuuren et al., 2011) is used for the
anthropogenic and biomass burning emissions during the years 1970-2020. Direct emissions of aerosol components from
biofuel and open biomass burning are considered by using scaling factors applied on the emitted black carbon based on the
findings of Akagi et al. (Akagi et al., 2011) (Table S1). Dust emission fluxes and emissions of crustal species ($Ca^{2+}$, $Mg^{2+}$, $K^+$,
$Na^+$) are calculated online as described by Klingmuller, et al. (Klingmuller et al., 2018) and based on the chemical composition
of the emitted soil particles in every grid cell (Karydis et al., 2016); Table S2. $NO_x$ produced by lightning is calculated online
and distributed vertically based on the parameterization of Grewe, et al. (Grewe et al., 2001). The emissions of NO from soils
are calculated online based on the algorithm of Yienger and Levy (Yienger and Levy, 1995). The oceanic DMS emissions are
calculated online by the AIRSEA Submodel (Pozzer et al., 2006). The natural emissions of $NH_3$ are based on the GEIA
database (Bouwman et al., 1997). Emissions of sea spray aerosols (assuming a composition suggested by Seinfeld and Pandis
(Seinfeld and Pandis, 2006); Table S1) and volcanic degassing emissions of $SO_2$ are based on the offline emission data set of
AEROCOM (Dentener et al., 2006).

## 4.3 Thermodynamic model

The inorganic aerosol composition, which is of prime importance for the accurate pH calculation, is computed with the
ISORROPIA-II thermodynamic equilibrium model (Fountoukis and Nenes, 2007). ISORROPIA-II calculates the
gas/liquid/solid equilibrium partitioning of the $K^+$-$Ca^{2+}$-$Mg^{2+}$-$NH_4^+$-$Na^+$-$SO_4^{2-}$-$NO_3^-$-$Cl^-$-$H_2O$ aerosol system and considers the
presence of 15 aqueous phase components and 19 salts in the solid phase. ISORROPIA-II solves for the equilibrium state by
considering the chemical potential of the species and minimizes the number of equations and iterations required by considering
specific compositional "regimes". The assumption of thermodynamic equilibrium is a good approximation for fine-mode
aerosols that rapidly reach equilibrium. However, the equilibrium timescale for large particles is typically larger than the time
step of the model (Meng and Seinfeld, 1996) leading to errors in the size distribution of semi-volatile ions like nitrate. Since
the current study include reactions of nitric acid with coarse sea-salt and dust aerosol cations, the competition of fine and
coarse particles for the available nitric acid can only be accurately represented by taking into account the kinetic limitations
during condensation of $HNO_3$ in the coarse mode aerosols. To account for kinetic limitations by mass transfer and transport
between the gas and particle phases, the process of gas/aerosol partitioning is calculated in two stages (Pringle et al., 2010).
First, the gaseous species that kinetically condense onto the aerosol phase within the model timestep are calculated assuming
diffusion limited condensation (Vignati et al., 2004). Then, ISORROPIA-II re-distributes the mass between the gas and the
aerosol phase assuming instant equilibrium between the two phases.
ISORROPIA-II is used in the forward mode, in which the total (i.e., gas and aerosol) concentrations are given as input.
Reverse mode calculations (i.e. when only the aerosol phase composition is known) should be avoided since they are sensitive
to errors and infer bimodal behaviour with highly acidic or highly alkaline particles, depending on whether anions or cations
are in excess (Song et al., 2018). While it is often assumed that aerosols are in a metastable state (i.e., composed only of a
supersaturated aqueous phase), here we use ISORROPIA-II in the thermodynamically stable state mode where salts are
allowed to precipitate once the aqueous phase becomes saturated. For this purpose, we have used the revised ISORROPIA-II
model which includes modifications proposed by Song et al. (2018), who resolved coding errors related to pH calculations
when the stable state assumption is used. By comparing with the benchmark thermodynamic model E-AIM, Song et al. (2018)
found that ISORROPIA-II produces somewhat higher pH (by 0.1-0.7 units, negatively correlated with RH). However, E-AIM
model versions either lack crustal cations from the ambient mixture of components (e.g. version II) (Clegg et al., 1998), or
only include $Na^+$ with the restriction that it should be used when RH> 60% (e.g. version IV) (Friese and Ebel, 2010). Song et
al. (2018) applied the revised ISORROPIA-II during winter haze events in eastern China and found that the assumed particle
phase state, either stable or metastable, does not significantly impact the pH predictions.
We performed a sensitivity simulation with only liquid particles (i.e., metastable), which revealed that the assumed particle
phase state does not significantly impact the pH calculations over oceans and polluted regions (e.g., Europe), however, the
metastable assumption produces more acidic particles (up to 2 units of pH) in regions affected by high concentrations of crustal
cations and consistently low RH values (Fig. S3). Fountoukis et al. (2007) have shown that the metastable solution predicts
significant amounts of water below the mutual deliquescence relative humidity (MDRH, where all salts are simultaneously
saturated with respect to all components). Further, the generally high calcium concentrations downwind of deserts results in
increasing pH values due to the precipitation of insoluble salts such as the $CaSO_4$. The metastable state assumption fails to
reproduce this since it treats only the ions in the aqueous phase. In general, high amounts of crustal species can significantly
increase the aerosol particle pH which is consistent with the presence of excess carbonate in the particle phase (Meng et al.,
1995). It is worth mentioning that the stable state solution algorithm of ISORROPIA II starts with assuming a dry aerosol, and
based on the ambient RH dissolves each of the salts depending on their DRH. However, in the ambient atmosphere, when the
RH over a wet particle is decreasing, it may not crystallize below the MDRH but instead remain in a metastable state affecting
the uptake of water by the particle and thus the pH. This could be the case in some locations with high diurnal variations of
RH. Our sensitivity calculations show that, overall, the stable state assumption produces an about 0.5 units higher global
average pH than the metastable assumption. Karydis et al. (2016) have shown that while the aerosol state assumption has a
marginal effect on the calculated nitrate aerosol tropospheric burden (2% change), it can be important over and downwind of
deserts at very low RHs where nitrate is reduced by up to 60% by using the metastable assumption. This is in accord with the
findings of Ansari and Pandis (2000) who suggested that the stable state results in higher concentrations of aerosol nitrate
when the RH is low (<35 %) and/or sulfate to nitrate molar ratios are low (<0.25).

**4.4 pH calculations**

The pH is defined as the negative decimal logarithm of the hydrogen ion activity ($a_{H^+} = \gamma x_{H^+}$) in a solution:

$$pH = -\log_{10}\left(\gamma x_{H^+}\right) \quad (A1)$$

where $x_{H^+}$ is the molality of hydrogen ions in the solution and $\gamma$ is the ion activity coefficient of hydrogen. Assuming that $\gamma$
is unity, the aerosol particle pH can be calculated by using the hydrogen ion concentration in the aqueous particle phase
calculated by ISORROPIA-II (in mole m$^{-3}$) and the aerosol water content calculated by GMXe (in mole Kg$^{-1}$). GMXe assumes
that particle modes are internally mixed and takes into account the contribution of both inorganic and organic (based on the
organic hygroscopicity parameter, kappa=0.14 (Tsimpidi et al., 2014)) species to aerosol water.
The aerosol particle pH is calculated online at each timestep, and output stored every five hours based on instantaneous
concentrations of fine aerosol water and hydrogen ions. The average pH values shown in the manuscript are based on the
calculated instantaneous mean pH values. According to the Jensen's inequality (Jensen, 1906), the average of the instantaneous
pH values is less than or equal to the pH calculated based on the average of the water and hydrogen ion instantaneous values.
We estimate that the average pH calculated based on 5-hourly instantaneous values is approximately 1-3 (~2 globally averaged)
units higher than the pH calculated based on the average water and hydrogen ion concentrations. By including online gas-
particle partitioning calculations of the NH$_3$/HNO$_3$ system in polluted air, as applied here, we find that the aerosol particle pH
is higher by approximately one unit (Guo et al., 2015). Hence by neglecting these aspects the aerosol particle pH would be
low-biased by about 3 units.

**4.5 Comparison against pH estimations from field derived PM$_{2.5}$ compositional data**

The pH calculated here is compared against pH estimations from field derived PM$_{2.5}$ compositional data around the world
compiled by Pye et al. (2020) (Table A1). pH data derived from other particle sizes (e.g., PM$_1$) has been omitted since aerosol
particle acidity can vary significantly with size (Zakoura et al., 2020). It should be emphasized that the comparison presented
in Table A1 aims to corroborate the spatial variability of pH found in this study and not to evaluate the model calculations.
Since direct measurements of aerosol particle acidity are not available, the observation-based aerosol particle pH is estimated
by employing thermodynamic equilibrium models (e.g., ISORROPIA) and making assumptions that can significantly affect
the results, especially when the data are averaged over extended periods, while RH conditions during data collection are not
always accounted for, e.g. in studies based on filter sampling. The calculation of aerosol particle acidity on a global scale

requires the advanced treatment of atmospheric aerosol chemical complexity, representing the real atmosphere, and beyond the conventional methods used by chemistry-climate models (CCM). The atmospheric chemistry model system EMAC is an ideal tool for this purpose since it is one of the most comprehensive CCM containing advanced descriptions of the aerosol thermodynamics (including e.g. dust-pollution interactions) and organic aerosol formation and atmospheric aging (affecting the aerosol water). Our model calculations for aerosol particle acidity are based on some processes/factors that are not included explicitly, usually neglected by model calculations used to constrain the aerosol particle acidity from observations. Sources of discrepancy between the pH calculations can be the following:

- The stable/metastable assumption does not affect the pH most of the time, however, in some cases with low RHs and the presence of crustal cations, the metastable assumption results in lower pHs (see section 4.3).

- Crustal species from deserts and $Na^+$ from sea salt can elevate the pH significantly in some locations, however, these are often neglected in observations.

- The organic aerosols (which are treated comprehensively by our model using the module ORACLE and the volatility basis set framework (Tsimpidi et al., 2014)) can contribute significantly to the aerosol water, and thus increase the aerosol particle pH. This contribution is not considered by many observational studies.

- Including gas phase species (e.g., $NH_3$, $HNO_3$) in the pH calculations is important. Using only the aerosol-phase as input (i.e., reverse mode) the inferred pH exhibits bimodal behaviour with very acidic or alkaline values depending on whether anions or cations are in excess (Hennigan et al., 2015). Even if the forward mode is used (without gas phase input), the calculated aerosol particle pH is biased low (approximately 1 pH unit) due to the repartition of semi-volatile anions (i.e., $NH_3$) to the gas phase to establish equilibrium (Guo et al., 2015).

- Another important aspect, not explicitly mentioned in many studies, relates to the methods used to derive the campaign-average (or for 3D models the simulated average) pH. In our model the aerosol particle pH is calculated online (2-minute time resolution), while output is stored every five hours based on instantaneous concentrations of fine particle $H_2O$ and $H^+$. This mimics 5-hourly aerosol sampling. Then, the average pH values are calculated from the instantaneous mean pH values (see section 4.4). Often models use average values (and not instantaneous) as output, or field-derived pH calculations use average observed $H_2O$ and $H^+$ values, which can result in important underestimation (by ~ 1-3 units) of the aerosol particle pH (Jensen, 1906).

- Some unrealistically high pH values in a few past studies resulted from coding errors in the stable state assumption of the ISORROPIA II model, which have been corrected in our study following the recommendation of Song et al. (2018).

- The type of thermodynamic model used is also important. Song et al. (2018) found that ISORROPIA-II produces somewhat higher pH (by 0.1-0.7 units, negatively correlated with RH) compared to the thermodynamic model E-AIM, which is used to observationally-constrain pH in some studies.

- Measurements of $PM_{2.5}$ nitrate are not always reliable because of artifacts associated with the volatility of ammonium nitrate (Schaap et al., 2004). Ammonium and nitrate can partially evaporate from Teflon filters at temperatures between

15 to 20 °C and can evaporate completely at temperatures above. The evaporation from quartz filters is also significant
at temperatures higher than 20 °C. This systematic underestimation of ammonium nitrate can affect the observed chemical
composition of the aerosol and thus the pH calculations.
• The comparison between global model output and observations at specific locations. This also concerns the aerosol
concentrations but is especially important for the aerosol particle acidity. Apart from the size of the model grid cells (i.e.,
~ 1.9°x1.9°), the altitude is also important. The first vertical layer of EMAC is approximately 67m in height. On the other
hand, ground observations are typically collected in a height up to 3 m. While the aerosol particles within size modes
simulated in our model are well-mixed, perhaps this is not the case for the aerosol particles observed at the surface and
potentially close to sources, and thus the aerosol particle acidity may be higher (e.g., due to the higher contribution from
local primary sources like $SO_4^{-2}$, lower water amounts in the aerosol, or lower concentrations of semi-volatile cations like
$NH_4^+$)

**4.6 Partitioning of nitric acid between the gas and aerosol phases**
The impact of pH on the fraction of nitrate in the particle phase relative to total nitrate (gas plus particle), i.e., $\varepsilon(NO_3^-)$, during
the 50 years of simulation in specific regions is calculated as follows (Nah et al., 2018):
$$\varepsilon(NO_3^-) = \frac{H_{HNO_3}^* WRT(0.987 \times 10^{-14})}{\gamma_{NO_3^-}\gamma_{H^+}10^{-pH} + H_{HNO_3}^* WRT(0.987 \times 10^{-14})} \quad (A2)$$

Where $H_{HNO_3}^*$ is the combined molality-based equilibrium constant of $HNO_3$ dissolution and deprotonation, $\gamma$'s represent the
activity coefficients, W is the aerosol water, R is the gas constant, and T is the ambient temperature. Eq. A2 is equivalent with
the instantaneous calculations of ISOROPIA II within EMAC. However, the model output is produced after considering all
processes in the model and is not calculated at every timestep. Therefore, the use of Eq. 2 can provide a clearer picture of the
impact of pH on $HNO_3$ gas/particle partitioning since the model output (e.g., gas-phase $HNO_3$ and nitrate in 4 size modes) is
subject to uncertainties related to other processes (e.g., deposition, coagulation, transport, etc.).
**4.7 Sulfate formation in aqueous aerosols**
The sulfate production rate on aqueous particles from the heterogeneous oxidation of S(IV) with the dissolved $O_3$ is given by
$$R_0 = k\,[O_3] \quad (A3)$$

. The first-order uptake rate, $k$, from monodisperse aerosols with radius $r_a$ and total aerosol surface $A$, is calculated following
Jacob (Jacob, 2000):

$$k = \left(\frac{r_\alpha}{D_g} + \frac{4}{\upsilon\gamma}\right)^{-1} A \quad (A4)$$

where $v$ is the mean molecular speed of $O_3$ and $D_g$ is its gas-phase molecular diffusion coefficient calculated as follows:
$$D_g = \frac{9.45 \times 10^{17} \times \sqrt{T \left(3.47 \times 10^{-2} + \frac{1}{M}\right)}}{\rho_{air}} \quad (A5)$$

where $T$ is the ambient air temperature, $\rho_{air}$ is the air density, and $M$ the molar mass of $O_3$. $\gamma$ is the reaction probability calculated
following Jacob (Jacob, 2000) and Shao et al. (Shao et al., 2019).
$$\gamma = \left(\frac{1}{\alpha} + \frac{v}{4HRT\sqrt{D_a K}}\frac{1}{f_r}\right) \quad (A6)$$

where α is the mass accommodation coefficient, Da is the aqueous-phase molecular diffusion coefficient of O3, H is the
effective Henry's law constant of $O_3$ (Sander, 2015), R is the ideal gas constant, $f_r$ is the reacto-diffusive correction term (Shao
et al., 2019), and K is the pseudo-first order reaction rate constant between S(IV) and $O_3$ in the aqueous phase (Seinfeld and
Pandis, 2006).

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

**Author contributions:** V.A.K. and J.L. planned the research, V.A.K., A.P.T. and A.P. performed the model calculations, V.A.K., A.P., and J.L. analyzed the results, V.A.K. and J.L. wrote the paper. All authors contributed to the manuscript.; **Competing interests:** Authors declare no competing interests. **Code/Data availability:** Data and related material can be obtained from V.A.K. (v.karydis@fz-juelich.de) upon request. **Acknowledgments:** The authors gratefully acknowledge the computing time granted on the supercomputer GAIA at Max Planck Institute for Chemistry, Mainz, and on the supercomputer JURECA through JARA at Forschungszentrum Jülich. The work of V.A.K. is supported by the European Union via its Horizon 2020 project FORCeS (GA 821205).

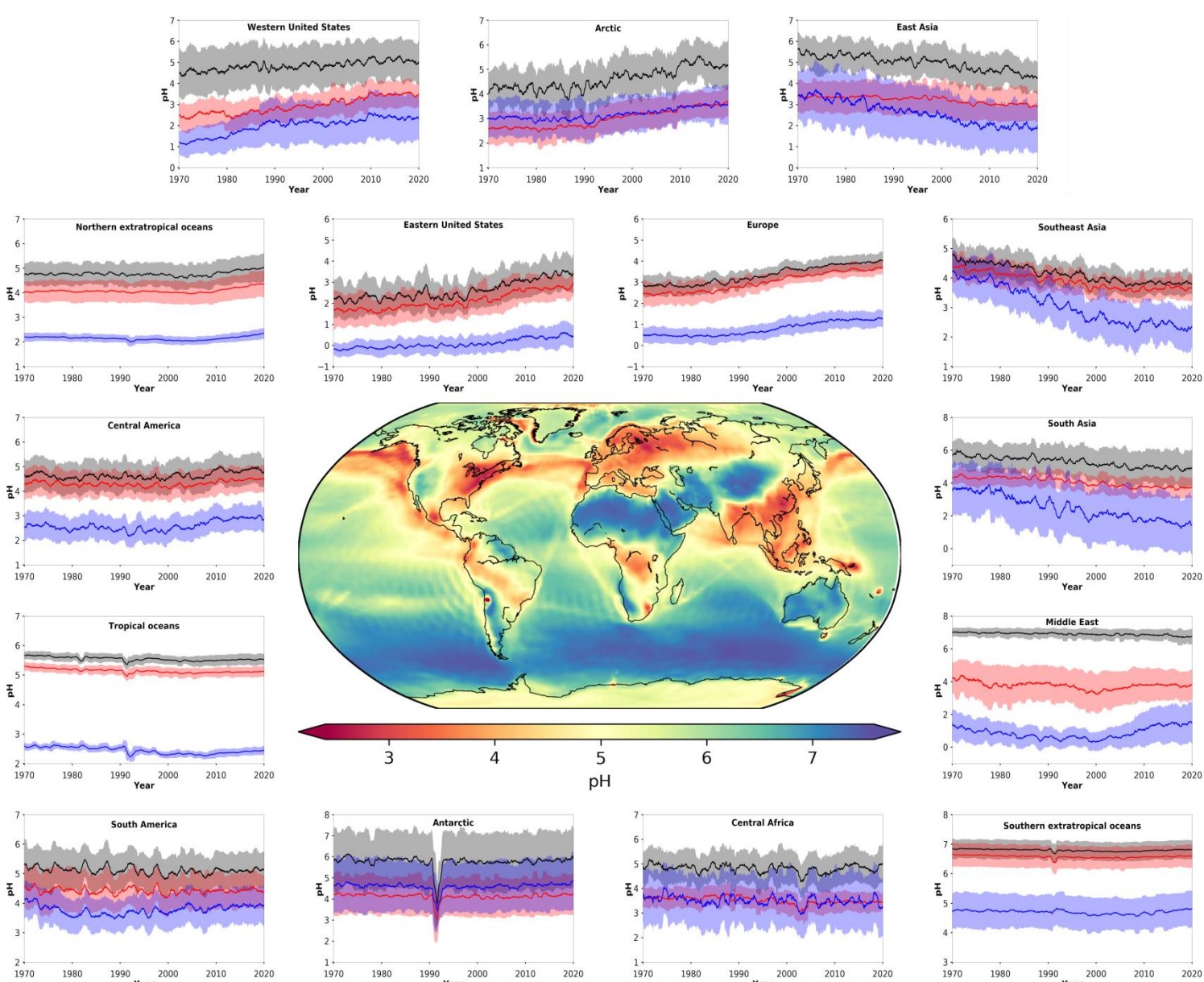

**Figure 1: Mean, near-surface fine aerosol particle pH during the period 2010-2015 (central panel). Surrounding panels show the temporal pH evolution during the period 1970-2020 at locations defined in Table 1. Black lines represent the reference simulation. Red and blue lines show the sensitivity simulations in which crustal particle and NH₃ emissions are removed, respectively. Ranges represent the 1σ standard deviation. The anomaly in 1991/2 is related to the Mt Pinatubo eruption.**

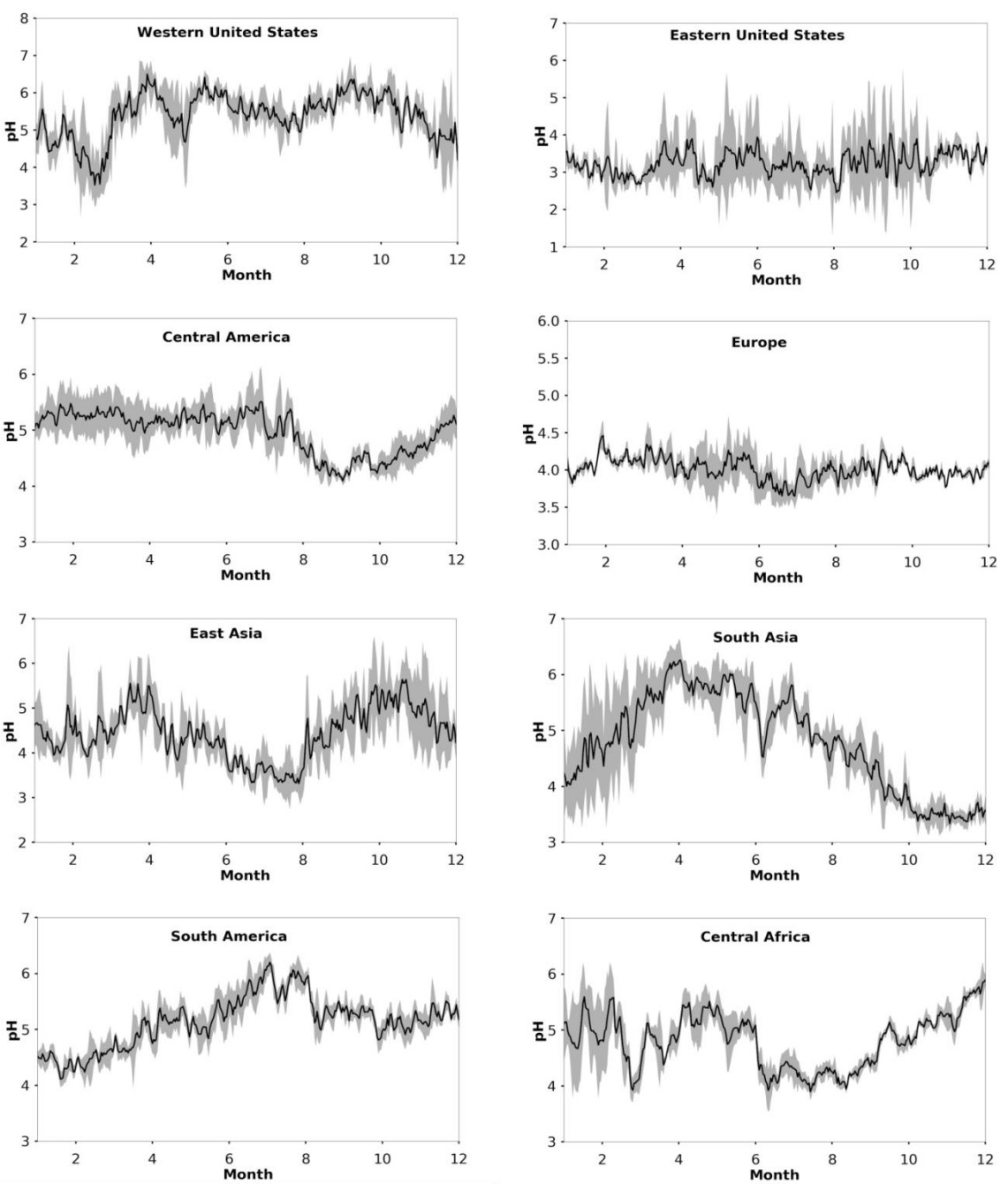

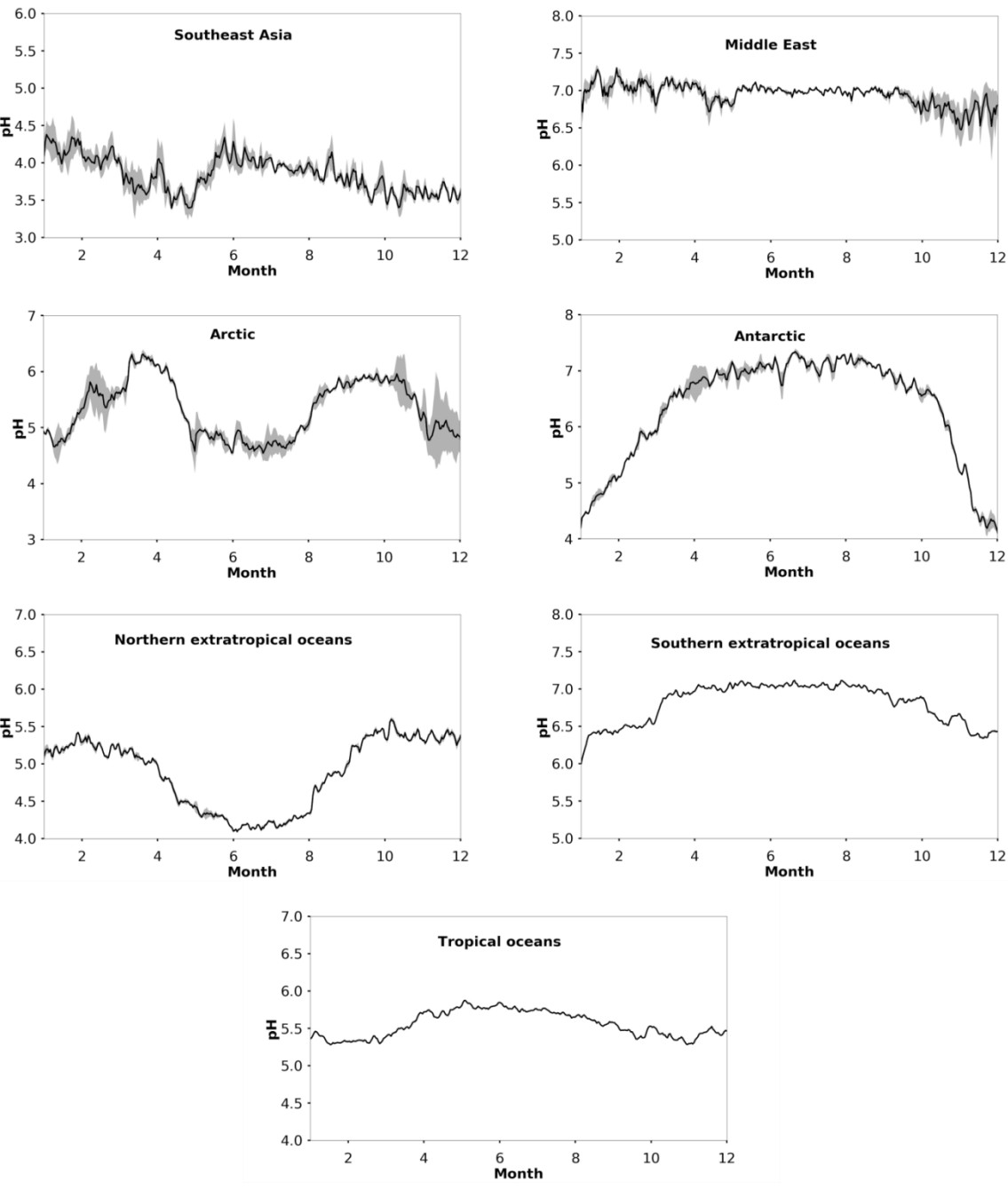

**Figure 2: Average seasonal cycle of modelled pH during the period 2010-2015 at locations defined in Table 1. Ranges represent the 1σ standard deviation.**

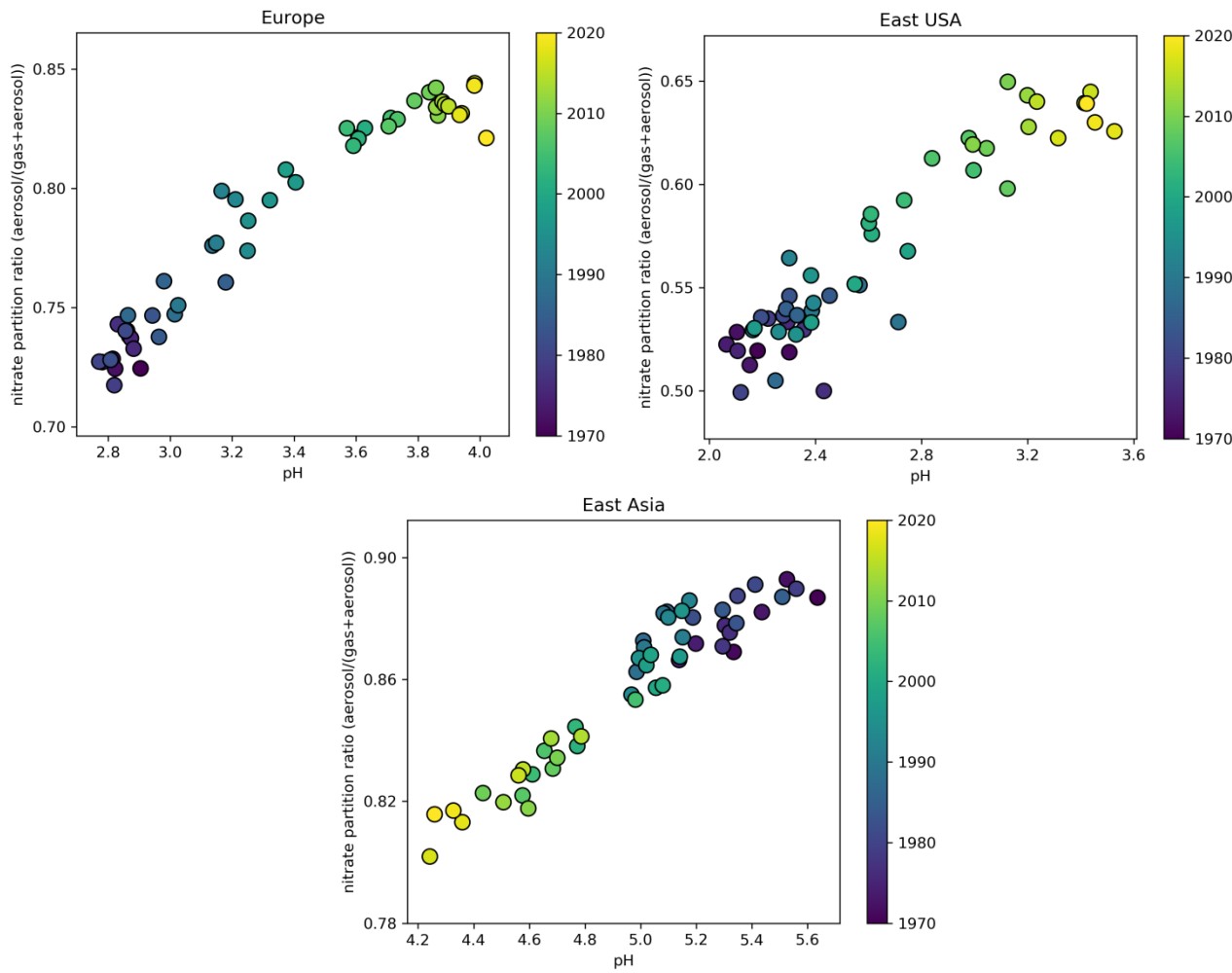

**Figure 3: Time evolution of particle phase fraction of total nitrate as a function of pH over Europe (left), the Eastern USA (right) and East Asia (bottom) during the period 1970-2020.**

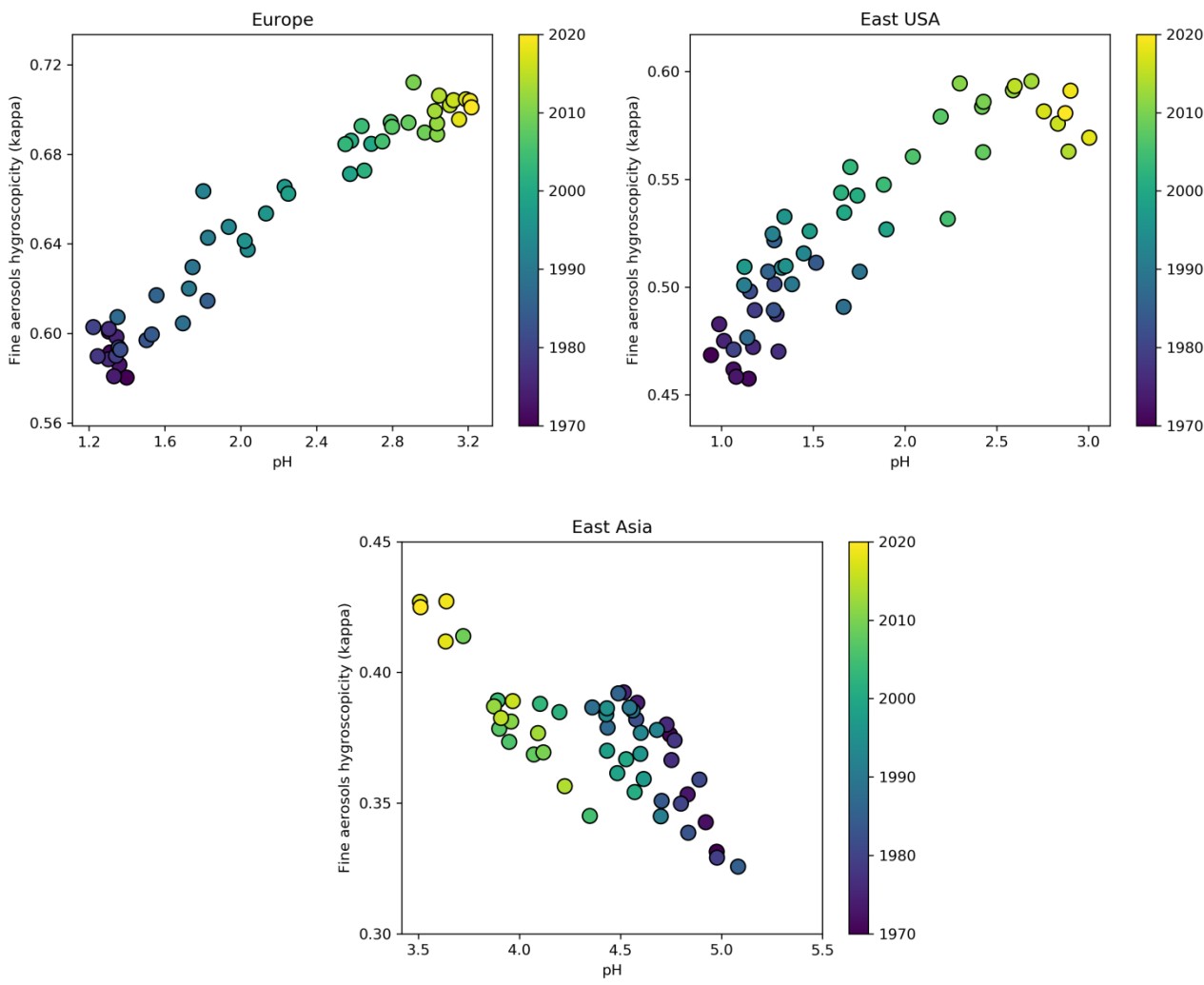

**Figure 4: Time evolution of annual average aerosol hygroscopicity (Kappa) as a function of pH over Europe (left), the Eastern USA (right) and East Asia (bottom) during the period 1970-2020 at the lowest cloud-forming level (940 hPa).**

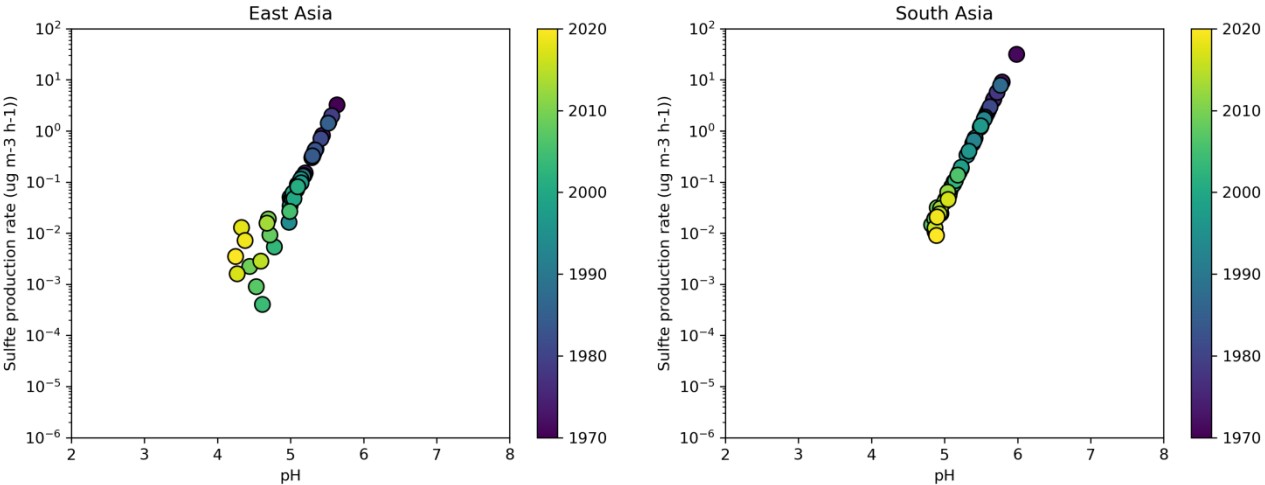

**Figure 5: Time evolution of the sulfate production rate on aqueous particles from the $SO_2+O_3$ multiphase chemistry reaction as a function of aerosol particle pH over East Asia (left) and South Asia (right) during the period 1970-2020.**

**Table 1: Decadal averages of aerosol particle pH.**

| Region | Longitude | Latitude | 1971-1980 | 1981-1990 | 1991-2000 | 2001-2010 | 2011-2020 |
|---|---|---|---|---|---|---|---|
| Western USA[1] | 90º-70ºW | 30º-46ºN | 4.6 | 4.8 | 4.8 | 5.0 | 5.1 |
| Eastern USA[1] | 124º-114ºW | 30º-52ºN | 2.2 | 2.4 | 2.4 | 2.9 | 3.3 |
| Central America[1] | 106º-52ºW | 4º-28ºN | 4.6 | 4.6 | 4.6 | 4.7 | 4.9 |
| Europe[1] | 12ºW-36ºE | 34º-62ºN | 2.8 | 3.0 | 3.3 | 3.7 | 3.9 |
| East Asia[1] | 100º-114ºE | 20º-44ºN | 5.3 | 5.2 | 5.1 | 4.7 | 4.5 |
| South Asia[1] | 68º-94ºE | 8º-32ºN | 5.6 | 5.5 | 5.3 | 5.0 | 4.9 |
| South America[1] | 75º-35ºW | 30º-0ºS | 5.2 | 5.1 | 5.1 | 5.1 | 5.1 |
| Central Africa[1] | 10º-40ºE | 10ºS-10ºN | 4.9 | 4.8 | 4.8 | 4.7 | 4.9 |
| Southeast Asia[1] | 94º-130ºE | 12ºS-20ºN | 4.5 | 4.3 | 4.1 | 3.9 | 3.8 |
| Middle East[1] | 36º-60ºE | 12º-34ºN | 7.0 | 7.0 | 6.9 | 6.9 | 6.8 |
| Arctic | 0º-360º | 60º-90ºN | 4.2 | 4.2 | 4.6 | 4.8 | 5.2 |
| North extratropics[2] | 0º-360º | 20º-60ºN | 4.8 | 4.8 | 4.7 | 4.7 | 4.9 |
| Tropical oceans[2] | 0º-360º | 20ºS-20ºN | 5.6 | 5.6 | 5.5 | 5.5 | 5.5 |
| South extratropics[2] | 0º-360º | 60º-20ºS | 6.8 | 6.8 | 6.8 | 6.8 | 6.8 |
| Antarctic | 0º-360º | 90º-60ºS | 5.9 | 5.9 | 5.6 | 5.8 | 5.8 |

[1]Only values over land are considered for the calculation of pH

[2]Only values over oceans are considered for the calculation of pH

**Table A1: Simulated fine aerosol particle pH compared to observationally-constrained estimates of fine particle acidity compiled by Pye et al. (2020).**

| Location | Latitude | Longitude | Time period | Simulated mean pH (Stable) | Simulated mean pH (Metastable) | Field derived mean pH | Method used | Reference |
|---|---|---|---|---|---|---|---|---|
| Pellston, MI, USA | 45.55°N | 84.78°W | Jul 2016 | 3.8 | 3.1 | 3.5 | pH indicator paper/ colorimetric image | Craig et al. (2018) |
| Ann Arbor, MI, USA | 42.28°N | 83.74°W | Aug 2016 | 4.3 | 3.0 | 3.5 | pH indicator paper/ colorimetric image | Craig et al. (2018) |
| Centreville, AL, USA | 32.9°N | 87.25°W | Jun 1998 – Aug 2013 | 6.4 | 5.7 | 1.2 | ISORROPIA (no $NH_3$) | Weber et al. (2016) |
| Centreville, AL, USA | 32.9°N | 87.25°W | Jun – Jul 2013 | 7.0 | 6.5 | 1.1 | ISORROPIA | Pye et al. (2018) |
| Egbert, ON, Canada | 44.23°N | 79.78°W | Jul – Sep 2012 | 3.9 | 3.5 | 2.1 | E-AIM Model II | Murphy et al. (2017) |
| Harrow, ON, Canada | 42.03°N | 82.89°W | Jun – Jul 2007 | 4.2 | 3.0 | 1.6 | E-AIM Model II | Murphy et al. (2017) |
| Pasadena, CA, USA | 34.14°N | 118.12°W | Jun 2010 | 5.9 | 2.7 | 2.7 | ISORROPIA (metastable) | Guo et al. (2017) |
| Toronto, Canada | 43.66°N | 79.40°W | 2007-2013 | 4.0 | 3.6 | 2.6 | E-AIM I (with gas $NH_3$, $HNO_3$) | Tao and Murphy (2019) |
| Toronto, Canada | 43.66°N | 79.40°W | 2014-2016 | 4.1 | 3.7 | 2.7 | E-AIM I (with gas $NH_3$, $HNO_3$) | Tao and Murphy (2019) |
| Ottawa, Canada | 45.43°N | 75.68°W | 2007-2016 | 4.0 | 3.9 | 2.5 | E-AIM I (with gas $NH_3$, $HNO_3$) | Tao and Murphy (2019) |
| Simcoe, Canada | 42.86°N | 80.27°W | 2007-2016 | 4.4 | 3.7 | 2.41 | E-AIM I (with gas $NH_3$, $HNO_3$) | Tao and Murphy (2019) |
| Montreal, Canada | 45.65°N | 73.57°W | 2007-2016 | 4.0 | 3.9 | 2.4 | E-AIM I (with gas $NH_3$, $HNO_3$) | Tao and Murphy (2019) |
| Windsor, Canada | 42.29°N | 83.07°W | 2007-2010 | 4.4 | 3.6 | 2.1 | E-AIM I (with gas $NH_3$, $HNO_3$) | Tao and Murphy (2019) |
| Windsor, Canada | 42.29°N | 83.07°W | 2012-2016 | 4.5 | 3.7 | 2.4 | E-AIM I (with gas $NH_3$, $HNO_3$) | Tao and Murphy (2019) |
| St. Anicet, Canada | 45.12°N | 74.29°W | 2007-2016 | 4.0 | 3.9 | 2.5 | E-AIM I (with gas $NH_3$, $HNO_3$) | Tao and Murphy (2019) |
| Sao Paulo, Brazil | 23.55°S | 46.63°W | Aug – Sep 2012 | 6.2 | 6.1 | 4.8 | E-AIM | Vieira-Filho et al. (2016) |
| Po Valley, Italy | 45.40°N | 12.20°E | Mar 2009 – Jan 2010 | 4.5 | 3.6 | 3.1 | E-AIM Model IV | Squizzato et al. (2013) |

| Po Valley, Italy | 45.40°N | 12.20°E | Spring 2009 | 4.3 | 3.7 | 3.6 | E-AIM Model IV | Squizzato et al. (2013) |
|---|---|---|---|---|---|---|---|---|
| Po Valley, Italy | 45.40°N | 12.20°E | Summer 2009 | 4.8 | 3.0 | 2.3 | E-AIM Model IV | Squizzato et al. (2013) |
| Po Valley, Italy | 45.40°N | 12.20°E | Fall 2009 | 4.5 | 3.6 | 3 | E-AIM Model IV | Squizzato et al. (2013) |
| Po Valley, Italy | 45.40°N | 12.20°E | Winter 2009-2010 | 4.4 | 4.0 | 3.4 | E-AIM Model IV | Squizzato et al. (2013) |
| Po Valley, Italy | 45.40°N | 12.20°E | Winter 2012-2013 | 4.2 | 4.0 | 3.9 | ISORROPIA (metastable, no $NH_3$) | Masiol et al. (2020) |
| Po Valley, Italy | 45.40°N | 12.20°E | Spring 2012 | 4.1 | 3.1 | 2.3 | ISORROPIA (metastable, no $NH_3$) | Masiol et al. (2020) |
| Cabauw, Netherlands | 51.97°N | 4.93°E | Jul 2012 – Jun 2013 | 4.0 | 3.8 | 3.7 | ISORROPIA | Guo et al. (2018) |
| Cabauw, Netherlands | 51.97°N | 4.93°E | Jun – Aug 2013 | 3.6 | 3.4 | 3.3 | ISORROPIA | Guo et al. (2018) |
| Cabauw, Netherlands | 51.97°N | 4.93°E | Dec – Feb 2012 | 4.1 | 4.1 | 3.9 | ISORROPIA | Guo et al. (2018) |
| Beijing, China | 39.99°N | 116.30°E | Nov 2015 – Dec 2016 | 4.9 | 4.2 | 4.2 | ISORROPIA | Liu et al. (2017) |
| Guangzhou, China | 23.13°N | 113.26°E | Jul 2013 | 2.6 | 1.9 | 2.5 | E-AIM Model IV | Jia et al. (2018) |
| Beijing, China | 39.97°N | 116.37°E | Nov 2014 –Dec 2014 | 4.5 | 5.3 | 4.6 | ISORROPIA | Song et al. (2018) |
| Beijing, China | 40.41°N | 116.68°E | Oct 2014 – Jan 2015 | 5.6 | 4.9 | 4.7 | ISORROPIA (metastable) | He et al. (2018) |
| Beijing, China | 39.99°N | 116.31°E | Jan – Dec 2014 | 4.9 | 4.0 | 3.0 | ISORROPIA (metastable) | Tan et al. (2018) |
| Beijing, China | 39.99°N | 116.31°E | Winter 2014 | 5.5 | 4.4 | 4.1 | ISORROPIA (metastable) | Tan et al. (2018) |
| Beijing, China | 39.99°N | 116.31°E | Fall 2014 | 6.0 | 4.6 | 3.1 | ISORROPIA (metastable) | Tan et al. (2018) |
| Beijing, China | 39.99°N | 116.31°E | Spring 2014 | 5.4 | 4.5 | 2.1 | ISORROPIA (metastable) | Tan et al. (2018) |
| Beijing, China | 39.99°N | 116.31°E | Summer 2014 | 3.1 | 2.4 | 1.8 | ISORROPIA (metastable) | Tan et al. (2018) |
| Tianjin, China | 39.11°N | 117.16°E | Dec 2014 – Jun 2015 | 4.4 | 3.7 | 4.9 | ISORROPIA (metastable) | Shi et al. (2017) |
| Tianjin, | 39.11°N | 117.16°E | Aug 2015 | 1.4 | 1.2 | 3.4 | ISORROPIA | Shi et al. (2017) |

| Location | Latitude | Longitude | Period | | | | Method | Reference |
|---|---|---|---|---|---|---|---|---|
| China | | | | | | | (metastable) | |
| Beijing, China | 39.98°N | 116.28°E | Feb 2017 | 4.7 | 4.8 | 4.5 | ISORROPIA | Ding et al. (2019) |
| Beijing, China | 39.98°N | 116.28°E | Apr - May 2016 | 5.2 | 4.7 | 4.4 | ISORROPIA | Ding et al. (2019) |
| Beijing, China | 39.98°N | 116.28°E | Jul - Aug 2017 | 2.2 | 1.9 | 3.8 | ISORROPIA | Ding et al. (2019) |
| Beijing, China | 39.98°N | 116.28°E | Sep - Oct 2017 | 4.5 | 3.7 | 4.3 | ISORROPIA | Ding et al. (2019) |
| Guangzhou, China | 23.13°N | 113.26°E | Jul – Sep 2013 | 2.7 | 2.2 | 2.4 | E-AIM Model III | Jia et al. (2018) |
| Hohhot, China | 40.48°N | 111.41°E | Summer 2014 | 5.5 | 4.0 | 5 | ISORROPIA (metastable, no $NH_3$) | Wang et al., 2019 |
| Hohhot, China | 40.48°N | 111.41°E | Autumn 2014 | 6.8 | 5.3 | 5.3 | ISORROPIA (metastable, no $NH_3$) | Wang et al. (2019) |
| Hohhot, China | 40.48°N | 111.41°E | Winter 2014 | 5.8 | 5.0 | 5.7 | ISORROPIA (metastable, no $NH_3$) | Wang et al. (2019) |
| Hohhot, China | 40.48°N | 111.41°E | Spring 2015 | 6.1 | 5.1 | 6.1 | ISORROPIA (metastable, no $NH_3$) | Wang et al. (2019) |
| Hohhot, China | 40.48°N | 111.41°E | 2014 - 2015 | 6.2 | 5.0 | 5.6 | ISORROPIA (metastable, no $NH_3$) | Wang et al. (2019) |
| Beijing, China | 40.41°N | 116.68°E | Oct 2014 – Jan 2015 | 5.6 | 4.9 | 7.6 | ISORROPIA (stable state) | He et al. (2018) |
| Xi'an, China | 34.23°N | 108.89°E | Nov – Dec 2012 | 5.7 | 4.5 | 6.7 | ISORROPIA | Wang et al. (2016) |
| Beijing, China | 39.99°N | 116.30°E | Jan – Feb 2015 | 5.0 | 3.8 | 7.6 | ISORROPIA | Wang et al. (2016) |
| Beijing, China | 40.35°N | 116.30°E | Jun – Aug 2005 | 4.2 | 3.3 | 0.6 | E-AIM Model II (only aerosols) | Pathak et al. (2009) |
| Shanghai, China | 31.45°N | 121.10°E | May – Jun 2005 | 3.5 | 3.1 | 0.7 | E-AIM Model II (only aerosols) | Pathak et al. (2009) |
| Lanzhou, China | 36.13°N | 103.68°E | Jun – Jul 2006 | 6.8 | 5.1 | 0.6 | E-AIM Model II (only aerosols) | Pathak et al. (2009) |
| Beijing, China | 40.32°N | 116.32°E | Jan 2005 – Apr 2006 | 5.1 | 4.1 | 0.7 | E-AIM Model II (only aerosols) | He et al. (2012) |
| Chongqing, China | 29.57°N | 106.53°E | Jan 2005 – Apr 2006 | 3.6 | 2.7 | 1.5 | E-AIM Model II (only aerosols) | He et al. (2012) |
| Beijing, China | 40°N | 116.33°E | Jan 2013 | 4.6 | 4.5 | 5.8 | ISORROPIA (forward & reverse, estimated | Wang et al. (2016) |

| | | | | | | (NH$_3$) | | |
|---|---|---|---|---|---|---|---|---|
| Singapore | 1.3$^o$N | 103.78$^o$E | Sep – Nov 2011 | 3.2 | 3.0 | 0.6 | E-AIM Model IV | Behera et al. (2013) |
| Hong Kong | 22.34$^o$N | 114.26$^o$E | Jul 1997 – May 1998 | 3.3 | 3.0 | 0.3 | E-AIM Model II (for RH >= 70%) | Yao et al. (2007) |
| Hong Kong | 22.34$^o$N | 114.26$^o$E | Nov 1996 – Nov 1997 | 3.4 | 2.9 | -1 | E-AIM Model II (for RH < 70%) | Yao et al. (2007) |
| Hong Kong | 22.34$^o$N | 114.26$^o$E | Oct 2008 | 5.0 | 3.2 | 0.6 | E-AIM Model III (only aerosols) | Xue et al. (2011) |
| Hong Kong | 22.34$^o$N | 114.26$^o$E | Nov 2008 | 3.7 | 2.7 | -0.5 | E-AIM Model III (only aerosols) | Xue et al. (2011) |
| Hong Kong | 22.34$^o$N | 114.26$^o$E | Jun - Jul 2009 | 1.6 | 2.0 | -0.1 | E-AIM Model III (only aerosols) | Xue et al. (2011) |
| Pacific Ocean | 47.5$^o$S | 147.5$^o$E | Nov - Dec 1995 | 7.0 | 6.5 | 1.0 | EQUISOLV | Fridlind and Jacobson (2000) |
| South Ocean | 61$^o$S | 45$^o$W | Jan 2015 | 6.9 | 6.7 | 1.4 | ISORROPIA (no NH$_3$) | Dall'Osto et al. (2019) |
| South Ocean | 64$^o$S | 65$^o$W | Jan – Feb 2015 | 6.9 | 6.8 | 3.8 | ISORROPIA (no NH$_3$) | Dall'Osto et al. (2019) |