# Peer review of "How alkaline compounds control atmospheric aerosol particle acidity"

_Atmospheric Chemistry and Physics, 2020_

## Referee Comment (RC1) · Anonymous Referee #1 · 19 Feb 2021

This paper from Karydis et al. predicts fine particle acidity, which is an important aerosol property linked to many particulate physicochemical processes, on the global scale and over a long historic period of 50 years. It discovers some interesting long-term trends in particle acidity with discussions on seasonal variabilities. Most importantly, it highlights the important roles of alkaline salts, such as ammonium and crustal cations, to buffer and elevate the global pH. The results are of interest to the geoscience community and supported by high-quality modeling, thus suitable for the scope of ACP Letter. However, several issues should be clarified before acceptance for publication, especially the large discrepancies in pH prediction (in some cases more than 2 units) when compared to observationally-constrained pH in previously reported studies, since the overestimation of pH results in exaggerating the importance of alkaline salts and

the accuracy of pH prediction determines its implications to atmospheric chemistry.

Major comments:

- Line 50&Line 218: the assumption of aerosol mode (solid+liquid vs. liquid) matters for pH prediction. For instance, it changes the estimated pH by more than 3 units in Pasadena. The current text in the method section lacks the explanation why the stable mode was chosen over metastable mode. More discussions would be useful to validate the model results. The Pasadena pH estimation in Guo et al. (2017) assumes metastable aerosols due to the high RH observed in that study (79 ± 17%). Considering the even higher RH after sunset, particles are highly likely to get deliquesced and stay so even in the daytime when RH drops below DRH (deliquescence relative humidity) but above ERH (efflorescence relative humidity). Such an effect would be observed in a place with a similar RH diurnal cycle. I wonder if a better way to present the model results is to choose the metastable mode for high RH cases/regions (such as the average RH of 60% and with nighttime/max RH over mutual DRH) and the stable mode for low RH cases/regions, especially when the two results deviate from each other by more than one pH unit. But the key judgment is which particle-phase assumption works the best to predict gas-particle partitioning of semi-volatile species comparing to observations (while the particle phase measurement/modeling is not available on the global scale). - Table S1 summarizes the comparison of simulated fine particle pH in this study to observationally-constrained pH in previous studies. In most cases, the simulated pH is higher, and the differences range from sub-one units up to six units. It is acknowledged that some previous estimations are biased low for lack of gas-phase input (e.g., Line 76). However, large differences are seen when compared to some observationally-constrained pH with gas-phase input, such as Pye et al. (2018) (7.0 vs. 1.1) and Murphy et al. (2017) (4.2 vs. 1.6). Also, the prediction of remote air in the Atlantic and Pacific Oceans (roughly 5-6 in Figure 1 other than lower values of 3 predicted for the northern parts that are heavily affected by anthropogenic emissions) is much higher than pH estimations based on ATom aircraft studies (roughly 0-1) (Nault et

al., 2020). Although the ATom estimations are based on submicron particles (and this study focuses on PM2.5), it is hard to believe that sea salts or mineral dust between 1 and 2.5 $\mu$m can elevate particle pH by 4-7 units on average. In summary, the results are very different, such as nearly neutral vs. highly acidic fine particles (i.e., very different implications for chemistry), requiring more discussions on the causes (e.g., crustal elements, mixing state, or particle phase). Possibilities include 1) that the simulated crustal elements may be externally mixed with sulfate/nitrate/ammonium aerosols or 2) the overestimation of crustal elements or the sum of ammonia and ammonium. In either case, the effects of alkaline compounds on the global fine particle acidity would be less than proposed. One way to tell the key factor(s) is by comparing the thermodynamic model inputs between the simulated ones and the field observations and do some sensitivity tests. - Caution should be paid towards Ca (especially for the cases of high Ca mass concentrations) due to the precipitation of CaSO4 as ISORROPIA-II assumes it to be completely insoluble. Some sensitivity tests may be carried out such as done in Kakavas et al. (2021). - Line 26: Please elaborate on how the cited papers show the effects of aerosol acidity on particle hygroscopic growth and its lifetime. The three papers talk about the importance of mineral dust in thermodynamic modeling. For example, Karydis et al. (2016) highlight that the tropospheric nitrate burden increases by 44% when considering dust aerosol chemistry but the connection between aerosol acidity to hygroscopicity or lifetime seems to be buried. - Line 66: "The aerosol pH over the anthropogenically-influenced northern hemispheric mid-latitudes exhibits a clear seasonal pattern with lower values during boreal summer and higher ones during winter, driven by the availability of ammonium and by the aerosol water content (Fig. 2)." First, please specify the locations after "northern hemispheric mid-latitudes". Second, it is not clear these regions exhibit clear seasonal variations as stated. For instance, the curves of the eastern US and Europe are nearly flat throughout the year, while the western US shows lower pH in the winter months (e.g., Dec, Jan, and Feb), opposite to the trends stated in the text. - Line 101: Suggest rephrasing the sentence as "Over North America, aerosol acidity also decreased with reduced SO2 and NOx

emissions." However, it seems to be more complicated for NOx than SO2, since more total nitrate may increase pH given the same amount of sulfate, transferring aerosols from a more acidic ammonium sulfate (or ammonium bisulfate) system to a less acidic ammonium nitrate system. - Line 118: The dominant H2O2 pathway at pH < 5 is for cloud droplets, not fine particles. Cheng et al. (2016) state that the NO2 pathway dominates at pH > 5 and the TMI pathway (transition metal ions) dominates at pH < 4.5 for the Beijing haze conditions. So even if the authors chose to only calculate the H2O2 pathway (which is probably the most important one for the less polluted cases at pH < 5) for the past 50 years, it is worth mentioning the other possible dominant pathway. - Line 131: Stating that NH3 is a major buffer is reasonable since it is often found in both gas and particle phases. Thus, it can redistribute between the two phases to buffer the pH. It remains to be explained though if the crustal elements simply increase particle pH or buffer the pH since they are non-volatile. For instance, although carbonate or bicarbonate is not considered in the ISORROPIA-II calculation, it could be the anion paired with crustal elements to buffer high pH for the H2CO3 pKa of 6.4 (The pKa of HCO3- is 10.3, which is too high to buffer the predicted pH predicted in this study). - Line 211: It would be useful to specify if the kinetic limitations affect simulations in this study and by what extent. The thermodynamic simulations based on observations often don't find the signs of kinetic limitations for fine particles (i.e., the predicted gas-particle partitioning agrees with observations, e.g. (Guo et al., 2017; Liu et al., 2017)), unless very fresh aerosols are sampled near the sources. - Line 267: It is not clear why Equation A2 is used to investigate the impact of pH on nitrate partitioning but not the results directly from ISORROPIA. The two should be equivalent. Please explain.

Minor comments:

- Line 31: Consider deleting "In the past" and changing the past form to present form since the ion balance and molar ratio methods still have these limitations and also don't consider the partial disassociation of acids, which could be added here. - Line 56: Change "high pH's are found…" to "high pH are found". - Line 78: Add "(Fig.

1)" after the sentence "Over the Arctic and the northern Atlantic and Pacific Oceans, aerosol acidity is significantly enhanced by strong sulfur emissions from international shipping and pollution transport from industrialized areas." Since the former and latter sentences are talking about Fig. 2. - Line 90: Does it make sense to have the most points in Fig. S1 with larger than one cation/anion ratios? Not for liquid only particles but reasonable for solid+liquid aerosols. So it would be great to explain this better either in the main text or in Fig. S1 caption. - Line 96: Provide kappa for ammonium sulfate and ammonium nitrate. - Fig. 1 caption: Add "during the period 1970-2020" after "Surrounding panels show the temporal pH evolution at locations defined in Table 1" to specify the time range (although it can be easily told from the panels). - Line 126: do you mean "overestimate"? Since the SO2 emission reduces drastically in Asia, the inventories are not updated in time to catch the reductions. Therefore, I would think overestimation makes more sense here logically. - Line 128: consider change "the large SO2 trends" to "the significant SO2 reduction trends" or "the long-term SO2 trends". - Line 154: add "is" after "NH3" to be "NH3 is also proved to be..." - Line 237: consider adding a reference for $\kappa$org = 0.14. Also, while the Greek alphabet of $\kappa$ is used here, "kappa" is used in Fig. 4. Better to be consistent.

References:

Cheng, Y., Zheng, G., Wei, C., Mu, Q., Zheng, B., Wang, Z., Gao, M., Zhang, Q., He, K., Carmichael, G., Poschl, U., and Su, H.: Reactive nitrogen chemistry in aerosol water as a source of sulfate during haze events in China, Sci. Adv., 2, e1601530, doi: 10.1126/sciadv.1601530, 2016. Guo, H., Liu, J., Froyd, K. D., Roberts, J. M., Veres, P. R., Hayes, P. L., Jimenez, J. L., Nenes, A., and Weber, R. J.: Fine particle pH and gas–particle phase partitioning of inorganic species in Pasadena, California, during the 2010 CalNex campaign, Atmos. Chem. Phys., 17, 5703-5719, doi: 10.5194/acp-17-5703-2017, 2017. Kakavas, S., Patoulias, D., Zakoura, M., Nenes, A., and Pandis, S. N.: Size-resolved aerosol pH over Europe during summer, Atmos. Chem. Phys., 21, 799-811, doi: 10.5194/acp-21-799-2021, 2021. Karydis, V.

A., Tsimpidi, A. P., Pozzer, A., Astitha, M., and Lelieveld, J.: Effects of mineral dust on global atmospheric nitrate concentrations, Atmos. Chem. Phys., 16, 1491-1509, doi: 10.5194/acp-16-1491-2016, 2016. Liu, M., Song, Y., Zhou, T., Xu, Z., Yan, C., Zheng, M., Wu, Z., Hu, M., Wu, Y., and Zhu, T.: Fine particle pH during severe haze episodes in northern China, Geophys. Res. Lett., 44, 5213-5221, doi: 10.1002/2017gl073210, 2017. Murphy, J. G., Gregoire, P. K., Tevlin, A. G., Wentworth, G. R., Ellis, R. A., Markovic, M. Z., and VandenBoer, T. C.: Observational constraints on particle acidity using measurements and modelling of particles and gases, Faraday Discuss., 200, 379-395, doi: 10.1039/c7fd00086c, 2017. Nault, B. A., Campuzano-Jost, P., Day, D. A., Jo, D. S., Schroder, J. C., Allen, H. M., et al.: Models underestimate the increase of acidity with remoteness biasing radiative impact calculations, AGU Fall Meeting, doi: https://agu.confex.com/agu/fm20/meetingapp.cgi/Paper/746418, 2020. Pye, H. O. T., Zuend, A., Fry, J. L., Isaacman-VanWertz, G., Capps, S. L., Appel, K. W., Foroutan, H., Xu, L., Ng, N. L., and Goldstein, A. H.: Coupling of organic and inorganic aerosol systems and the effect on gas-particle partitioning in the southeastern US, Atmos. Chem. Phys., 18, 357-370, doi: 10.5194/acp-18-357-2018, 2018.

---

## Referee Comment (RC2) · Anonymous Referee #2 · 1 Jun 2021

This paper uses a model to predict fine particle (PM2.5) pH globally. They find more acidic particles in the more anthropogenically-influenced regions and basic particles in regions of high non-volatile cations, a finding that is not highly surprising but which does provide a general verification of the method. Their major finding is on how alkaline compounds control PM2.5 particle acidity and these trends over the past 50 years.

The devil is in the details and this is especially true when assessing aerosol particle pH and particle pH impacts. As noted by the 1st reviewer, the pH predicted by the model is off by a wide margin in some locations relative to predictions supported by data. I would note that the model is often significantly off in locations where the pH predictions have been assessed through comparisons between observed gas/particle

partitioning of HNO3 and NH3 to predicted values and where partitioning of at least of these species is sensitive to pH, meaning there is high confidence in the pH reported for these cases. The first reviewer provided significant details on this issue. I will not repeat those suggestions and instead look a a broader view.

I calculate that the mean (median) pH difference (simulated – field derived) from the data provided in Table S1 is 1.61 (1.4), suggesting the model is systematically predicting a high pH globally (the authors may wish to check my calculations).

I suggest the authors spend more time on first making sure, and discussing in more detail, the quality of the pH predictions. What causes this high pH bias compared to other reported studies and what are the implications. A greater focus on this apparent discrepancy is important since this manuscript is based only on a model prediction and incorrectly predicted pH has significant ramifications. First, a major finding reported is on the role of alkaline species that raises the particle pH; a high bias pH would indicate that the role of alkaline species is overstated in this analyses. Second, the paper also focuses on the partitioning of HNO3, which is highly non-linear with pH, where HNO3 can change from all in the gas phase to all in the particle phase over a change in pH of about 1 to 2 units, near the level of the mean difference found in the comparison, as noted above. Thus the bias could have a large impact on this finding as well. Overall, it is not clear what new contribution this paper makes on understanding aerosol pH. Substantial modification based on a better assessment of the model should be required prior to consideration for publication.

Aside, I do not see the seasonality in mid N American latitudes (noted in lines 67-68, Fig 2), which also seems to disagree with two independent observational studies (Wong et al, 2020; Tao et al, 2019) and which has significant implications.

Tao, Y., and J. G. Murphy. 2019. 'The sensitivity of PM2.5 acidity to meteorological parameters and chemical composition changes: 10-year records from six Canadian monitoring sites', Atm. Chem. Phys., 19: 9309-20.

Wong, J. P. S., Y. Yang, T. Fang, J. A. Mulholland, A. Russell, S. Ebelt, A. Nenes, and R. J. Weber. 2020. 'Fine particle iron in soils and road dust is modulated by coal-fired power plant sulfur', Envir. Sci Technol., 54: 7088-96.

---

## Author Comment (AC1) · 7 Jul 2021

*This paper from Karydis et al. predicts fine particle acidity, which is an important aerosol property linked to many particulate physicochemical processes, on the global scale and over a long historic period of 50 years. It discovers some interesting longterm trends in particle acidity with discussions on seasonal variabilities. Most importantly, it highlights the important roles of alkaline salts, such as ammonium and crustal cations, to buffer and elevate the global pH. The results are of interest to the geoscience community and supported by high-quality modeling, thus suitable for the scope of ACP Letter. However, several issues should be clarified before acceptance for publication, especially the large discrepancies in pH prediction (in some cases more than 2 units) when compared to observationally-constrained pH in previously reported studies, since the overestimation of pH results in exaggerating the importance of alkaline salts and the accuracy of pH prediction determines its implications to atmospheric chemistry.*

We would like to thank the reviewer for his/her positive response and for the very thoughtful review. By raising important issues, the reviewer helped us further to better present our results and improve the manuscript. Below is a point by point response on the comments and suggestions.

**Major comments**

*1. Line 50 & Line 218: the assumption of aerosol mode (solid+liquid vs. liquid) matters for pH prediction. For instance, it changes the estimated pH by more than 3 units in Pasadena. The current text in the method section lacks the explanation why the stable mode was chosen over metastable mode. More discussions would be useful to validate the model results. The Pasadena pH estimation in Guo et al. (2017) assumes metastable aerosols due to the high RH observed in that study (79 ± 17%). Considering the even higher RH after sunset, particles are highly likely to get deliquesced and stay so even in the daytime when RH drops below DRH (deliquescence relative humidity) but above ERH (efflorescence relative humidity). Such an effect would be observed in a place with a similar RH diurnal cycle. I wonder if a better way to present the model results is to choose the metastable mode for high RH cases/regions (such as the average RH of 60% and with nighttime/max RH over mutual DRH) and the stable mode for low RH cases/regions, especially when the two results deviate from each other by more than one pH unit. But the key judgment is which particle-phase assumption works the best to predict gas-particle partitioning of semi-volatile species comparing to observations (while the particle phase measurement/modeling is not available on the global scale).*

We agree with the reviewer that the aerosol state assumption is important for the pH calculations with ISORROPIA and needs further attention in our manuscript. Here we used a revised ISORROPIA-II model which includes modifications proposed by Song et al. (2018), who resolved coding errors related to pH calculations when the stable state assumption is used. Applying the revised model during winter haze events in Beijing, they have found that the assumed particle phase state, either stable or metastable, does not significantly impact the pH predictions. However, a sensitivity simulation in our study (e.g., applying both stable and metastable assumptions on a global scale) revealed that even if the assumed particle phase state does not significantly impact the pH calculations over oceans and polluted regions (i.e., characterized by high RH values), the metastable assumption produces more acidic particles (up to 2 units of pH) in regions affected by

high concentrations of crustal cations (i.e., downwind of desert areas) and consistently low RH values. Fountoukis et al. (2007) have shown that the metastable solution predicts significant amounts of water below the mutual DRH (MDRH, where all salts are simultaneously saturated with respect to all components). In addition, the presence of high calcium concentrations downwind of the deserts results in increasing pH values due to the missing precipitation of insoluble salts such as the $CaSO_4$. This is something that the metastable state assumption fails to reproduce since it treats only the ions in the aqueous phase. In general, high amounts of crustal species can significantly increase the aerosol pH which is consistent with the presence of excess carbonate in the aerosol phase (Meng et al., 1995). Since our model is applied on a global scale, we believe that the stable state assumption can reproduce the sensitivities of pH more accurately given that the focus of our manuscript is on insights regarding the impacts of alkaline species on aerosol acidity. The stable state assumption gives almost identical results with the metastable in areas with high $NH_3$ concentrations (e.g., over central Europe) and at the same time is more appropriate for regions affected by crustal elements (i.e., close to deserts). Overall, the stable state assumption used here as a basecase simulation produces about 0.5 units higher global average pH than the metastable assumption.

Polluted areas that are downwind of crustal sources (like Pasadena) are of special interest and will be discussed more elaborately in the manuscript. When calculating species concentrations, the stable state solution algorithm of ISORROPIA II starts with assuming a completely dry aerosol and based on the ambient RH dissolves each of the salts depending on their DRH. However, in the ambient atmosphere, when the RH over a wet particle is decreasing, the wet aerosol may not crystallize below the MDRH but instead remain in a metastable state affecting the uptake of water by the aerosol and thus the pH. We agree with the reviewer that this can happen in some locations with high diurnal variations of RH. However, over Pasadena, the observed RH is always high (87±9%) for the period used for our model comparison (second half of the campaign where $PM_{2.5}$ were measured). Our sensitivity analysis revealed that the aerosol state was not affected by the state assumption since over Pasadena, both stable and metastable assumptions predict the same amount of water in the aerosol. We believe that the differences on pH are due to the high concentrations of calcium from the Great Basin Desert which results in the precipitation of high amounts of $CaSO_4$, lowering the particle acidity (but without affecting the water activity since $CaSO_4$ is insoluble and does not contribute to the MDRH depression). It is worth mentioning that calcium was not included in the Guo et al. (2017) study which can explain the differences in the observed and simulated aerosol acidity. Our sensitivity analysis shows that the simulated particle-phase fraction of nitrate over Pasadena is 40% using the stable state assumption and 32% using the metastable assumption, compared to the observed 51%. This is in accordance with the findings of Ansari and Pandis (2000) who suggested that the stable state results in higher concentrations of aerosol nitrate when the RH is low (<35 %) and/or sulfate to nitrate molar ratios are low (<0.25). Karydis et al. (2016) have shown that while the aerosol state assumption has a marginal effect on the calculated nitrate aerosol tropospheric burden (2% change), it can be important over deserts at very low RHs where nitrate is reduced by up to 60% by using the metastable assumption.

*2.    Table S1 summarizes the comparison of simulated fine particle pH in this study to observationally-constrained pH in previous studies. In most cases, the simulated pH is higher, and the differences range from sub-one units up to six units. It is acknowledged that some previous estimations are biased low for lack of gas-phase input (e.g., Line 76). However, large differences*

*are seen when compared to some observationally-constrained pH with gas-phase input, such as Pye et al. (2018) (7.0 vs. 1.1) and Murphy et al. (2017) (4.2 vs. 1.6). Also, the prediction of remote air in the Atlantic and Pacific Oceans (roughly 5-6 in Figure 1 other than lower values of 3 predicted for the northern parts that are heavily affected by anthropogenic emissions) is much higher than pH estimations based on ATom aircraft studies (roughly 0-1) (Nault et al., 2020). Although the ATom estimations are based on submicron particles (and this study focuses on PM2.5), it is hard to believe that sea salts or mineral dust between 1 and 2.5 μm can elevate particle pH by 4-7 units on average. In summary, the results are very different, such as nearly neutral vs. highly acidic fine particles (i.e., very different implications for chemistry), requiring more discussions on the causes (e.g., crustal elements, mixing state, or particle phase). Possibilities include 1) that the simulated crustal elements may be externally mixed with sulfate/nitrate/ammonium aerosols or 2) the overestimation of crustal elements or the sum of ammonia and ammonium. In either case, the effects of alkaline compounds on the global fine particle acidity would be less than proposed. One way to tell the key factor(s) is by comparing the thermodynamic model inputs between the simulated ones and the field observations and do some sensitivity tests.*

The calculation of aerosol acidity on a global scale requires the advanced treatment of atmospheric aerosol chemical complexity, analogous to the real atmosphere and beyond the conventional methods used by the current chemistry-climate models (CCM). The atmospheric chemistry model system EMAC is the ideal tool for this purpose since it is one of the most comprehensive CCM containing advanced descriptions of the aerosol thermodynamics (including the dust-pollution interactions) and organic aerosol formation and atmospheric aging (affecting the aerosol water). Therefore, the comprehensive global atmospheric multiphase chemistry simulations of the past 50 years presented in this study enabled us for the first time to provide accurate aerosol acidity calculations and the associated sensitivities. Our model calculations for aerosol acidity are based on several important processes/factors that are not included explicitly, or usually neglected, by model calculations used to constrain the aerosol acidity from observations. These are the following:

1/ As discussed above, the stable/metastable assumption does not affect the simulated pH most of the time, however, in some case with low RHs and the presence of crustal cations, the metastable assumption results in lower pHs.

2/ Crustal species from surrounding deserts and $Na^+$ from sea salt can elevate the pH significantly in some locations, however, these are often neglected from observations.

3/ The organic aerosols (which are treated comprehensively by our model using the module ORACLE and the volatility basis set framework) can contribute significantly to the aerosol water, and thus increase the aerosol pH. This contribution is not considered by many observational studies.

4/ The inclusion of gas phase species (e.g., $NH_3$, $HNO_3$) in the pH calculations is important, since using only the aerosol-phase as input (i.e., reverse mode) the inferred pH exhibits a bimodal behavior with very acidic or alkaline values depending on whether anions or cations are in excess (Hennigan et al., 2015). Even if the forward mode is used (without gas phase input), the calculated aerosol pH is biased low (approximately 1 unit of pH) due to the repartition of semivolatile anions (i.e., $NH_3$) to the gas phase to establish equilibrium (Guo et al., 2015).

5/ Another important aspect, usually not mentioned in many studies, relates to the methods used to derive the campaign-average (or for 3D models the simulated average) pH. In our model the aerosol pH is calculated online (2-minute time resolution), while output is stored every five hours based on instantaneous concentrations of fine aerosol $H_2O$ and $H^+$. According to Jensen's inequality (Jensen, 1906), the average of the instantaneous pH values is less than or equal to the pH calculated based on the average of the $H_2O$ and $H^+$ instantaneous values. We estimate that the average pH calculated based on 5-hourly instantaneous values is approximately 1-3 (~2 globally averaged) units higher than the pH calculated based on the average $H_2O$ and $H^+$ concentrations. If other models are using average values (and not instantaneous) as output, or if field-derived pH calculations are using average observed $H_2O$ and $H^+$ values, this can result in important underestimations of aerosol pH.

6/ Some unrealistically high pH values in a few past studies resulted from coding errors in the stable state assumption of ISORROPIA II model, which have been fixed in our study following the recommendation of Song et al. (2018).

7/ The type of thermodynamic model used is also important. Song et al. (2018) has found that ISORROPIA-II produces somewhat higher pH (by 0.1-0.7 units, negatively correlated with RH) compared to the thermodynamic model E-AIM, which is used to observationally-constrain pH in some studies.

8/ Measurements of PM2.5 nitrate are not always reliable because of artifacts associated with the volatility of ammonium nitrate (Schaap et al., 2004). Ammonium and nitrate can partially evaporate from Teflon filters at temperatures between 15 to 20 °C and can evaporate completely at temperatures above. The evaporation from quartz filters is also significant at temperatures higher than 20 °C. This systematic underestimation of ammonium nitrate can affect the observed chemical composition of the aerosol and thus the pH calculations.

9/ A final important issue refers to the comparison between global model output and observations at specific locations. This also concerns the aerosol concentrations but is especially important for a tentative property like the aerosol acidity. Apart from the size of our grid cells (which is $1.9^o$x$1.9^o$), the altitude is also important. Our model has a first layer which is approximately 67m in height. On the other hand, ground observations are typically collected in a height up to 3 m. While the aerosols within size modes simulated in our model are well-mixed, perhaps this is not the case for the aerosols observed so close to the surface and potentially to sources, and thus the aerosol acidity may be higher (e.g., due to the higher contribution from local primary sources like $SO_4^{-2}$, lower water in the aerosol, or lower semivolatile cations like $NH_4^+$)

All the above points are now extensively discussed in the manuscript. Concerning the comparison with observationally derived pH from aircraft campaigns, we need to emphasize that all pH values reported in our study are near the surface and cannot be directly compared to aircraft campaigns. Every aerosol size mode in our model is well mixed while the size modes are externally mixed. Therefore, we agree with the reviewer that our crustal elements are in many cases externally mixed with sulfates and ammonium but nitrates do exist in our large particles as well since they condense to the coarse (and accumulation) mode particles in order to maintain the charge balance in the aerosol phase. However, the aerosol pH of $PM_{2.5}$ is calculated based on the total $H_2O$ and $H^+$ in the aerosol until the cut-off point of 2.5 μm in our aerosol lognormal distributions. Our model predicts important amounts of crustals and $Na^+$ in the $PM_{1-2.5}$ size range, therefore the pH of $PM_{2.5}$ is meaningfully higher than that of $PM_1$. Finally, it is worth mentioning that Nault et al. (2020) did not include $Na^+$ in their suit of components.

*3.    Caution should be paid towards Ca (especially for the cases of high Ca mass concentrations) due to the precipitation of CaSO4 as ISORROPIA-II assumes it to be completely insoluble. Some sensitivity tests may be carried out such as done in Kakavas et al. (2021).*

This indeed is a critical point as discussed earlier in our response for the case of Pasadena. Calcium is the major crustal component of dust in most deserts (Karydis et al., 2016) and unlike Mg, K, and Na it can react with sulfate ions and form insoluble $CaSO_4$, which precipitates out of the aerosol aqueous phase. This interaction reduces the aqueous sulfate and thus the aerosol acidity. In our sensitivity simulation without crustal component emissions, this effect is evident in the western United States and East Asia where high concentrations of sulphate interact with strong emissions of calcium emitted from the Great Basin and the Gobi deserts, respectively, resulting in the increase of pH by up to two units (Figure 1 of the manuscript). The role of $CaSO_4$ in this sensitivity response is now emphasized in the revised text.

*4.    Line 26: Please elaborate on how the cited papers show the effects of aerosol acidity on particle hygroscopic growth and its lifetime. The three papers talk about the importance of mineral dust in thermodynamic modeling. For example, Karydis et al. (2016) highlight that the tropospheric nitrate burden increases by 44% when considering dust aerosol chemistry but the connection between aerosol acidity to hygroscopicity or lifetime seems to be buried.*

We agree with the reviewer that the Karydis et al. (2016) study focuses more on the thermodynamic interactions between mineral dust anions and inorganic cations and their impact on nitrate aerosol formation. In the revised text we have replaced this study with the Karydis et al. (2017) work where the impacts of these thermodynamic interactions on aerosol hygroscopicity and cloud droplet formation are revealed.

*5.    Line 66: "The aerosol pH over the anthropogenically-influenced northern hemispheric mid-latitudes exhibits a clear seasonal pattern with lower values during boreal summer and higher ones during winter, driven by the availability of ammonium and by the aerosol water content (Fig. 2)." First, please specify the locations after "northern hemispheric mid-latitudes". Second, it is not clear these regions exhibit clear seasonal variations as stated. For instance, the curves of the eastern US and Europe are nearly flat throughout the year, while the western US shows lower pH in the winter months (e.g., Dec, Jan, and Feb), opposite to the trends stated in the text*

This is correct. Not every region of the Northern Hemisphere exhibits this behavior. This is mostly evident over highly polluted regions like East Asia and not over Europe and Eastern USA. The Northern extratropical Oceans also show the same clear seasonal pattern. We have corrected the text accordingly.

*6.    Line 101: Suggest rephrasing the sentence as "Over North America, aerosol acidity also decreased with reduced SO2 and NOx emissions." However, it seems to be more complicated for NOx than SO2, since more total nitrate may increase pH given the same amount of sulfate, transferring aerosols from a more acidic ammonium sulfate (or ammonium bisulfate) system to a less acidic ammonium nitrate system.*

The SO$_2$ emissions over North America have decreased more steeply compared to the NO$_x$ emissions. However, even if this was not the case, the reduction of NO$_x$ alone cannot affect the sulfate concentrations (by replacing it with nitrates in the aerosol). The available sulfuric acid condenses instantaneously onto the aerosol phase and form ammonium sulfate (or ammonium bisulfate) and only then the nitric acid can form ammonium nitrate with the free NH$_3$ left.

*7.    Line 118: The dominant H2O2 pathway at pH < 5 is for cloud droplets, not fine particles. Cheng et al. (2016) state that the NO2 pathway dominates at pH > 5 and the TMI pathway (transition metal ions) dominates at pH < 4.5 for the Beijing haze conditions. So even if the authors chose to only calculate the H2O2 pathway (which is probably the most important one for the less polluted cases at pH < 5) for the past 50 years, it is worth mentioning the other possible dominant pathway.*

What we have calculated in our work is the O$_3$ pathway which is important for both cloud droplets and aerosols for pH above 5 units. However, we agree with the reviewer that the reference to the H$_2$O$_2$ pathway in the text is misleading, and now we refer to the other important SO$_2$ oxidation pathways in the aqueous aerosol phase (e.g., NO$_2$ above pH=5 and TMI below pH=5).

*8.    Line 131: Stating that NH3 is a major buffer is reasonable since it is often found in both gas and particle phases. Thus, it can redistribute between the two phases to buffer the pH. It remains to be explained though if the crustal elements simply increase particle pH or buffer the pH since they are non-volatile. For instance, although carbonate or bicarbonate is not considered in the ISORROPIA-II calculation, it could be the anion paired with crustal elements to buffer high pH for the H2CO3 pKa of 6.4 (The pKa of HCO3- is 10.3, which is too high to buffer the predicted pH predicted in this study).*

Overall, in the text we refer to the buffering capacity of crustal elements as a term to describe their activity to decrease the pH of the aerosol with respect to an increase in the acid concentrations (i.e., owing to the anthropogenic activities). We recognize that the term is not completely accurate since crustal cations are non-volatile and can only increase the pH (and not buffer it). Following the reviewer's comment, we have revised the text accordingly.

*9.    Line 211: It would be useful to specify if the kinetic limitations affect simulations in this study and by what extent. The thermodynamic simulations based on observations often don't find the signs of kinetic limitations for fine particles (i.e., the predicted gas-particle partitioning agrees with observations, e.g. (Guo et al., 2017; Liu et al., 2017)), unless very fresh aerosols are sampled near the sources.*

The assumption of thermodynamic equilibrium is a good approximation for fine-mode aerosols that can reach equilibrium very fast. However, the equilibrium timescale for large particles is typically larger than the time step of the model (Meng and Seinfeld, 1996) leading to errors in the size distribution of semi-volatile ions like nitrate. Since the current study include reactions of nitric acid with coarse sea-salt and dust aerosol cations, the competition of fine and coarse particles for the available nitric acid can only be accurately represented by taking into account the kinetic limitations during condensation of HNO$_3$ in the coarse mode aerosols. This information has been added to the text.

**10.** *Line 267: It is not clear why Equation A2 is used to investigate the impact of pH on nitrate partitioning but not the results directly from ISORROPIA. The two should be equivalent. Please explain.*

Indeed, equation A2 is in theory equivalent with the instant calculations of ISOROPIA II within our global model EMAC. However, the output of our model is not every timestep (is every 5 hours), and only after every other process in the model is calculated. Therefore, if we use the model output (e.g., gas-phase $HNO_3$ and $NO_3^-$ in 4 size modes) the result would be subject to uncertainties from other processes (e.g., deposition, coagulation, transport, etc.). The use of Eq. 2 can provide us a clearer picture of the impact of pH on $HNO_3$ gas/particle partitioning. We have added this information in the text.

**Minor comments**

**1.** *Line 31: Consider deleting "In the past" and changing the past form to present form since the ion balance and molar ratio methods still have these limitations and also don't consider the partial disassociation of acids, which could be added here.*

Thank you for pointing this out. We have revised the sentence accordingly.

**2.** *Line 56: Change "high pH's are found. . ." to "high pH are found".*

*Done.*

**3.** *Line 78: Add "(Fig. 1)" after the sentence "Over the Arctic and the northern Atlantic and Pacific Oceans, aerosol acidity is significantly enhanced by strong sulfur emissions from international shipping and pollution transport from industrialized areas." Since the former and latter sentences are talking about Fig. 2.*

Thank you for the suggestion.

**4.** *Line 90: Does it make sense to have the most points in Fig. S1 with larger than one cation/anion ratios? Not for liquid only particles but reasonable for solid+liquid aerosols. So it would be great to explain this better either in the main text or in Fig. S1 caption.*

Yes, the cation/anion ratio include all ions from both solid salts and the liquid phase. This is now explained in the text.

**5.** *Line 96: Provide kappa for ammonium sulfate and ammonium nitrate.*

The kappa hygroscopicity parameters for ammonium sulfate and ammonium nitrate are 0.53 and 0.67, respectively. The information has been added to the text.

*6.* *Fig. 1 caption: Add "during the period 1970-2020" after "Surrounding panels show the temporal pH evolution at locations defined in Table 1" to specify the time range (although it can be easily told from the panels).*

The sentence has been revised as "Surrounding panels show the temporal pH evolution during the period 1970-2020 at locations defined in Table 1."

*7.* *Line 126: do you mean "overestimate"? Since the SO2 emission reduces drastically in Asia, the inventories are not updated in time to catch the reductions. Therefore, I would think overestimation makes more sense here logically.*

We meant that inventories tend to underestimate the real trends (reductions) in emissions. Admittedly, this was quite confusing and now we have rewritten the sentence as "$SO_2$ emission trends since 2007 have been so drastic that inventories and scenarios tend to overestimate the emitted $SO_2$."

*8.* *Line 128: consider change "the large SO2 trends" to "the significant SO2 reduction trends" or "the long-term SO2 trends".*

Done.

*9.* *Line 154: add "is" after "NH3" to be "NH3 is also proved to be..."*

Done.

*10.* *Line 237: consider adding a reference for κorg = 0.14. Also, while the Greek alphabet of κ is used here, "kappa" is used in Fig. 4. Better to be consistent.*

Done.
**References:**

Ansari, A. S., and Pandis, S. N.: The effect of metastable equilibrium states on the partitioning of nitrate between the gas and aerosol phases, Atmospheric Environment, 34, 157-168, 10.1016/s1352-2310(99)00242-3, 2000.

Cheng, Y., Zheng, G., Wei, C., Mu, Q., Zheng, B., Wang, Z., Gao, M., Zhang, Q., He, K., Carmichael, G., Poschl, U., and Su, H.: Reactive nitrogen chemistry in aerosol water as a source

of sulfate during haze events in China, Sci. Adv., 2, e1601530, doi: 10.1126/sciadv.1601530, 2016.

Guo, H., Liu, J., Froyd, K. D., Roberts, J. M., Veres, P. R., Hayes, P. L., Jimenez, J. L., Nenes, A., and Weber, R. J.: Fine particle pH and gas–particle phase partitioning of inorganic species in Pasadena, California, during the 2010 CalNex campaign, Atmos. Chem. Phys., 17, 5703-5719, doi: 10.5194/acp-17-5703-2017, 2017.

Kakavas, S., Patoulias, D., Zakoura, M., Nenes, A., and Pandis, S. N.: Size-resolved aerosol pH over Europe during summer, Atmos. Chem. Phys., 21, 799-811, doi: 10.5194/acp-21-799-2021, 2021. Karydis, V. A., Tsimpidi, A. P., Pozzer, A., Astitha, M., and Lelieveld, J.: Effects of mineral dust on global atmospheric nitrate concentrations, Atmos. Chem. Phys., 16, 1491-1509, doi: 10.5194/acp-16-1491-2016, 2016.

Karydis, V. A., Tsimpidi, A. P., Pozzer, A., Astitha, M., and Lelieveld, J.: Effects of mineral dust on global atmospheric nitrate concentrations, Atmos. Chem. Phys., 16, 1491-1509, 10.5194/acp-16-1491-2016, 2016.

Karydis, V. A., Tsimpidi, A. P., Bacer, S., Pozzer, A., Nenes, A., and Lelieveld, J.: Global impact of mineral dust on cloud droplet number concentration, Atmospheric Chemistry and Physics, 17, 5601-5621, 10.5194/acp-17-5601-2017, 2017.

Liu, M., Song, Y., Zhou, T., Xu, Z., Yan, C., Zheng, M., Wu, Z., Hu, M., Wu, Y., and Zhu, T.: Fine particle pH during severe haze episodes in northern China, Geophys. Res. Lett., 44, 5213-5221, doi: 10.1002/2017gl073210, 2017.

Meng, Z. Y., Seinfeld, J. H., Saxena, P., and Kim, Y. P.: Atmospheric gas-aerosol equilibrium .4. Thermodynamics of carbonates, Aerosol Science and Technology, 23, 131-154, 1995.

Meng, Z. Y., and Seinfeld, J. H.: Time scales to achieve atmospheric gas-aerosol equilibrium for volatile species, Atmospheric Environment, 30, 2889-2900, 10.1016/1352-2310(95)00493-9, 1996.

Murphy, J. G., Gregoire, P. K., Tevlin, A. G., Wentworth, G. R., Ellis, R. A., Markovic, M. Z., and VandenBoer, T. C.: Observational constraints on particle acidity using measurements and modelling of particles and gases, Faraday Discuss., 200, 379-395, doi: 10.1039/c7fd00086c, 2017.

Nault, B. A., Campuzano-Jost, P., Day, D. A., Jo, D. S., Schroder, J. C., Allen, H. M., et al.: Models underestimate the increase of acidity with remoteness biasing radiative impact calculations, AGU Fall Meeting, doi: https://agu.confex.com/agu/fm20/meetingapp.cgi/Paper/746418, 2020.

Pye, H. O. T., Zuend, A., Fry, J. L., Isaacman-VanWertz, G., Capps, S. L., Appel, K. W., Foroutan, H., Xu, L., Ng, N. L., and Goldstein, A. H.: Coupling of organic and inorganic aerosol systems and the effect on gas-particle partitioning in the southeastern US, Atmos. Chem. Phys., 18, 357-370, doi: 10.5194/acp-18-357-2018, 2018.

Schaap, M., van Loon, M., ten Brink, H. M., Dentener, F. J., and Builtjes, P. J. H.: Secondary inorganic aerosol simulations for Europe with special attention to nitrate, Atmos. Chem. Phys., 4, 857-874, 10.5194/acp-4-857-2004, 2004.

Song, S., Gao, M., Xu, W., Shao, J., Shi, G., Wang, S., Wang, Y., Sun, Y., and McElroy, M. B.: Fine-particle pH for Beijing winter haze as inferred from different thermodynamic equilibrium models, Atmos. Chem. Phys., 18, 7423-7438, 10.5194/acp-18-7423-2018, 2018.

---

## Author Comment (AC2) · 7 Jul 2021

*This paper uses a model to predict fine particle (PM$_{2.5}$) pH globally. They find more acidic particles in the more anthropogenically-influenced regions and basic particles in regions of high non-volatile cations, a finding that is not highly surprising but which does provide a general verification of the method. Their major finding is on how alkaline compounds control PM2.5 particle acidity and these trends over the past 50 years.*

We thank the reviewer for his/her review of our manuscript and the helpful comments. Below is a point by point response to his/her comments.

*The devil is in the details and this is especially true when assessing aerosol particle pH and particle pH impacts. As noted by the 1st reviewer, the pH predicted by the model is off by a wide margin in some locations relative to predictions supported by data. I would note that the model is often significantly off in locations where the pH predictions have been assessed through comparisons between observed gas/particle partitioning of HNO3 and NH3 to predicted values and where partitioning of at least of these species is sensitive to pH, meaning there is high confidence in the pH reported for these cases. The first reviewer provided significant details on this issue. I will not repeat those suggestions and instead look a a broader view.*
*I calculate that the mean (median) pH difference (simulated – field derived) from the data provided in Table S1 is 1.61 (1.4), suggesting the model is systematically predicting a high pH globally (the authors may wish to check my calculations).*
*I suggest the authors spend more time on first making sure, and discussing in more detail, the quality of the pH predictions. What causes this high pH bias compared to other reported studies and what are the implications. A greater focus on this apparent discrepancy is important since this manuscript is based only on a model prediction and incorrectly predicted pH has significant ramifications. First, a major finding reported is on the role of alkaline species that raises the particle pH; a high bias pH would indicate that the role of alkaline species is overstated in this analysis. Second, the paper also focuses on the partitioning of HNO3, which is highly non-linear with pH, where HNO3 can change from all in the gas phase to all in the particle phase over a change in pH of about 1 to 2 units, near the level of the mean difference found in the comparison, as noted above. Thus, the bias could have a large impact on this finding as well. Overall, it is not clear what new contribution this paper makes on understanding aerosol pH. Substantial modification based on a better assessment of the model should be required prior to consideration for publication.*

As discussed in response to the second comment by the first reviewer, the calculation of aerosol acidity on a global scale requires the advanced treatment of atmospheric aerosol chemical complexity, analogous to the real atmosphere and beyond the conventional methods used by the current chemistry-climate models (CCM). The atmospheric chemistry model system EMAC is the ideal tool for this purpose since it is one of the most comprehensive CCM containing advanced descriptions of the aerosol thermodynamics (including the dust-pollution interactions) and organic aerosol formation and atmospheric aging (affecting the aerosol water). Therefore, the comprehensive global atmospheric multiphase chemistry simulations of the past 50 years presented in this study enabled us for the first time to provide advanced aerosol acidity calculations

and the associated sensitivities. Our model calculations for aerosol acidity are based on several important processes/factors that are not included explicitly, or usually neglected, by most of the model-calculations used to constrain the aerosol acidity from observations, and this leads to higher pH values in our analysis. In brief, these factors are the following and are further analyzed in our response to the first reviewer: 1/ the stable/metastable assumption, 2/ The lack of crustal species in the observations, 3/ the omission of the organic aerosols contribution to the aerosol water, 4/ The use of the reverse mode of ISORROPIA, or the lack of gas phase species (e.g., $NH_3$, $HNO_3$) in the pH calculations, 5/ Uncertainties on the methods used to derive the campaign-average (or for 3D models the simulated average) pH, 6/ Coding errors in the stable state assumption of ISORROPIA II model in past studies, 7/ The type of thermodynamic model used (e.g., E-AIM vs. ISORROPIA), 8/ Measurement artifacts associated with the volatilization of ammonium nitrate from filters, 9/ The mixing state of aerosol due to the location and height of observational samples compared to the well-mixed aerosols from our model with grid size $1.9^o$x$1.9^o$ and ~67m height.

*Aside, I do not see the seasonality in mid N American latitudes (noted in lines 67-68, Fig 2), which also seems to disagree with two independent observational studies (Wong et al, 2020; Tao et al, 2019) and which has significant implications.*

This is correct. A clear seasonal pattern is mostly evident over highly polluted regions like East Asia and not over Europe and Eastern USA. The Northern extratropical Oceans also exhibit seasonality. We have corrected the text accordingly.

*Tao, Y., and J. G. Murphy. 2019. 'The sensitivity of PM2.5 acidity to meteorological parameters and chemical composition changes: 10-year records from six Canadian monitoring sites', Atm. Chem. Phys., 19: 9309-20.*

*Wong, J. P. S., Y. Yang, T. Fang, J. A. Mulholland, A. Russell, S. Ebelt, A. Nenes, and R. J. Weber. 2020. 'Fine particle iron in soils and road dust is modulated by coal-fired power plant sulfur', Envir. Sci Technol., 54: 7088-96.*

---

## Author Response (AR2)

**Response to review by editor:**

*I am happy to accept your paper for final publication as a Letter in ACP subject to some relatively minor further changes:*

We would like to thank the editor for his time to edit the manuscript and his positive response for publishing our research as a Letter in ACP. Below is a point by point response on his comments and suggestions.

*1) Please consider the location and naming of Table 1/S1 and whether (as requested by the reviewer) it is possible to add some observational data. If you intend it to be in the appendix then please name it Table A1.*

The table that includes the comparison of simulated aerosol pH against the observationally-constrained estimates has been moved to the appendix. The second referee requested to highlight any field data in Table S1 that is not influenced by the bullet points of section 4.5. While this is a good suggestion in principle, this is not feasible since every single filed data has a few (or more) sources of discrepancy with the simulated pH as outlined by the listed bullet-points.

*2) You are not consistent in your use of the words aerosol, aerosols and particles. In most cases the word particle would be most appropriate (particles have acidity, not the aerosol). When referring to aerosol in multiple locations then the plural is acceptable, otherwise the word aerosols is not a synonym of particles. You also use the word particulate as an adjective in the abstract where it should be a noun (particle mass).*

Following the editor's recommendations, we have changed the word aerosol/aerosols to "aerosol particle acidity" and "aerosol pH" to "aerosol particle pH" throughout the manuscript.

*3) The final sentence of the abstract is somewhat vague. What control strategies are you referring to in the context of climate change? The abstract up to this point has been only about pH, so it is a large leap for readers to understand the link between pH and climate, which is presumably though particle hygroscopicity and size. Rather than being a speculative statement, please try to link this sentence more to the findings described in your paper. The conclusions could also explain this link in a bit more detail so that this sentence is linked to a longer description in the paper.*

We have revised the abstract and the conclusions in order to more clearly present the link between the aerosol particle acidity and climate, i.e., our findings revealed an increase in aerosol hygroscopicity following the simulated changes in aerosol particle pH.

---

## Author Response (AR3)

**Response to review by editor:**

*Before final acceptance I would like to make some additional recommendations based on the modifications you made in response to my comments.*

We would like to thank the editor for his thorough review. Below is a point-by-point response on his recommendations.

*1) Please correct the error "decreased" on line 181. You may want to double check that throughout the paper you do not have the same mistake in terms of increased acidity/decreased pH. It's easily done.*

The error in line 181 has been corrected. No other similar mistake found in the text.

*2) I think your abstract could summarise better the key result. I suggest to start a new sentence at "and uncovered remarkable variability..." and add something about the increases and decreases over Europe, N America and Asia. E.g., "The simulations uncover..." You may slightly exceed the 200 word limit, but please try to reduce back if feasible.*

We thank the reviewer for his recommendation to further improve our abstract. We have added one sentence to summarize the particle acidity changes in the three main polluted regions.

*3) Regarding "aerosol particle". You are mostly consistent now, but there are a few places where you talk about aerosol acidity (in the abstract) or aerosol in place of particle. I don't think it is necessary to say aerosol particle every time. Once you have introduced in the abstract/intro that you are discussing aerosol particles (rather than cloud particles) then it is sufficient to just refer to particle pH, particle acidity, etc. Sorry if my previous comment was not clear.*

We have thoroughly revised the text following the editor's recommendation.